# Ultraviolet light-induced collagen degradation inhibits melanoma invasion

Timothy Budden[1], Caroline Gaudy-Marqueste[2], Andrew Porter [3], Emily Kay[4,5], Shilpa Gurung[1], Charles H. Earnshaw [1], Katharina Roeck[1], Sarah Craig [1], Víctor Traves[6], Jean Krutmann [7,8], Patricia Muller[9], Luisa Motta[10], Sara Zanivan [4,5], Angeliki Malliri [3], Simon J. Furney [11,12], Eduardo Nagore [6] & Amaya Virós [1✉]

Ultraviolet radiation (UVR) damages the dermis and fibroblasts; and increases melanoma incidence. Fibroblasts and their matrix contribute to cancer, so we studied how UVR modifies dermal fibroblast function, the extracellular matrix (ECM) and melanoma invasion. We confirmed UVR-damaged fibroblasts persistently upregulate collagen-cleaving matrix metalloprotein-1 (*MMP1*) expression, reducing local collagen (*COL1A1*), and COL1A1 degradation by MMP1 decreased melanoma invasion. Conversely, inhibiting ECM degradation and MMP1 expression restored melanoma invasion. Primary cutaneous melanomas of aged humans show more cancer cells invade as single cells at the invasive front of melanomas expressing and depositing more collagen, and collagen and single melanoma cell invasion are robust predictors of poor melanoma-specific survival. Thus, primary melanomas arising over collagen-degraded skin are less invasive, and reduced invasion improves survival. However, melanoma-associated fibroblasts can restore invasion by increasing collagen synthesis. Finally, high *COL1A1* gene expression is a biomarker of poor outcome across a range of primary cancers.

[1] Skin Cancer and Ageing Lab, Cancer Research UK Manchester Institute, The University of Manchester, Manchester, UK. [2] Department of Dermatology and Skin Cancers, Aix-Marseille University, Marseille, France. [3] Cell Signalling Group, Cancer Research UK Manchester Institute, The University of Manchester, Alderley Park, Manchester, UK. [4] Institute of Cancer Sciences, University of Glasgow, Glasgow, UK. [5] CRUK Beatson Institute, Glasgow, UK. [6] Department of Dermatology, Institut Valencià Oncologia, Valencia, Spain. [7] IUF – Leibniz Research Institute of Environmental Medicine, Düsseldorf, Germany. [8] Medical Faculty, Heinrich-Heine-University, Düsseldorf, Germany. [9] Tumour Suppressors Lab, Cancer Research UK Manchester Institute, The University of Manchester, Manchester, UK. [10] Department of Histopathology, Salford Royal NHS Foundation Trust, The University of Manchester, Manchester, UK. [11] Genomic Oncology Research Group, Department of Physiology and Medical Physics, Royal College of Surgeons in, Ireland Dublin, Ireland. [12] Centre for Systems Medicine, Royal College of Surgeons in Ireland Dublin, Dublin, Ireland. ✉email: Amaya.viros@cruk.manchester.ac.uk

UVR is the major environmental risk factor for the development of melanoma[1–5] and sun exposure is the main cause of rising disease incidence[3]. While the association between UVR and melanoma incidence is well established[5], there are controversial clinical studies associating sun exposure, or sun damage to the dermis, with improved melanoma survival[6–8]. However, other studies have found no association between sun damage and outcome, and clinical studies show melanomas arising on the scalp and neck, areas likely chronically sun damaged, are linked to poor outcome[9–11].

UVR damage accumulates with increasing decades of life, and aged patients have worse melanoma survival[12–15]. Therefore, it is possible that chronic UVR damage may lead to shorter melanoma-specific survival (MSS). However, in common with some non-hormonal cancers, the incidence and mortality of melanoma sharply rise after age 60, and then significantly decrease after age 85 (refs. [16,17]), suggesting the relationship between cumulative UVR exposure, cutaneous damage, age and melanoma death is not linear.

Previous studies have shown collagen quantity in the extracellular matrix (ECM) modifies melanoma cell behaviour[18]. Surprisingly, both increased[19] and decreased[20] deposition of collagen have been linked to malignant behaviour, suggesting the effect of collagen on cancer behaviour extends beyond protein level and scaffold function. In this study, we explore how collagen levels in the dermis, which vary according to sun damage and age, affect melanoma survival.

## Results

### Somatic mutation burden in dermal fibroblasts correlates with extracellular matrix degradation and collagenase expression.

The pivotal task of dermal fibroblasts is to regulate ECM remodelling, including the turnover of collagen[21]. In chronically UVR-damaged skin there is an increase of mutations[22] and degraded collagen that is not compensated by new collagen synthesis, contributing to overall ECM degradation. We analysed gene expression in human adult fibroblasts to compare matched UVR-damaged and UVR-protected dermis from healthy donors (median age 42, range 19–66, ref. [23]). We used COSMIC total signature 7 mutations[24], which indicate UVR-induced damage, as a surrogate marker of accumulated UVR exposure. Strikingly, the most significantly differentially expressed pathway was the ECM pathway (Fig. 1a, b, Supplementary Data 1 and Supplementary Table 1), demonstrating progressive downregulation of collagen genes (Supplementary Table 2) and upregulation of matrix metalloproteinases (MMP), including matrix metalloproteinase-1 (*MMP1*), in UVR-damaged adult donor fibroblasts, with increasing signature 7 mutations. Furthermore, *MMP1* was highly expressed in donor fibroblasts from UVR-exposed calves of adults in the Genotype-Tissue Expression cohort[25] (median TPM = 51.85, $n = 504$, median all tissues TPM = 0.078).

MMP1 cleaves collagen 1 (COL1A1) after acute UVR exposure[26,27], so we compared the secretion of MMP1, *COL1A1* expression and ECM collagen deposition of isogenic UVR-naive fibroblasts from a human foreskin fibroblast cell line (HFF) and UVR-damaged fibroblasts (UV-HFF) 2 weeks after UVR exposure (Supplementary Fig. 1a). We found UV-HFF fibroblasts increased the secretion of MMP1 (Supplementary Fig. 1b) with no compensatory increase in *COL1A1* transcription (Supplementary Fig. 1c) or deposition in the ECM of UV-HFF, compared to UVR-naive fibroblasts (COL1A1 mean label-free quantification (LFQ) intensity HFF = 33.20, UV-HFF = 33.47, $q$ value = 0.16, COL1A2 HFF = 32.57, UV-HFF = 32.87, $q$ value = 0.08, Supplementary Fig. 1d). In addition, atomic force microscopy (AFM) topographic imaging suggested

UV-HFF fibroblast-generated ECM presented more fragmented, sparser and disorganised matrix fibrils than UVR-naive, HFF fibroblasts. The higher roughness (Rq) value indicates less symmetry across the ECM surface plane, in keeping with degradation of UV-HFF fibroblast-generated ECM[28–30] (Fig. 1c, d). Furthermore, immunofluorescent staining of fibronectin fibres in HFF and UV-HFF-derived ECM, confirmed that UV-HFF matrices were significantly more disorganised with fewer aligned fibres than matrices generated by HFF fibroblasts (Fig. 1e–g).

Since UVR damage alters fibroblast function, compromising ECM renewal, we compared the density of collagen fibres in chronically sun-damaged and sun-protected healthy skin of aged patients ($n = 16$, age > 59); confirming reduction of collagen in UVR-damaged dermis[31] (Fig. 1h). In addition, we confirmed fibroblasts from tumour-adjacent sun-damaged patient dermis (solar elastosis[31,32]) have higher total somatic mutation burden[22] ($n = 13$, Fig. 1i), indicating that cumulative UVR leads to dermal ECM degradation and decreased collagen.

### Low collagen concentration and reduced collagen integrity decrease melanoma invasion.

To study if UVR damage to fibroblasts driving collagen degradation affects melanoma progression, we compared melanoma invasion in spheroids embedded in matrices of increasing collagen concentrations. Melanoma cell lines present varying degrees of invasion, so we used three cell lines established from tumours bearing different UVR mutation signatures, reflecting UVR and non-UVR tumour origins[33] (Supplementary Fig. 2a). We found that regardless of the UVR history of the melanoma cell line, the invasion of the three melanoma lines was optimal in 1.5 mg/ml collagen, and higher (2.5 mg/ml, $p < 0.0001$) and lower collagen concentrations significantly reduced invasion into the ECM (0.25 mg/ml, $p < 0.0001$, 0.5 mg/ml, $p = 0.03$; Fig. 2a, b and Supplementary Fig. 2b). In addition, we quantified the number of melanoma cells detaching from the spheroid and invading as single cells, and found single cell invasion optimal within a range of collagen concentrations, decreasing with higher and lower collagen densities (Fig. 2c, d). We generated organotypic dermal constructs with HFF or UV-HFF foreskin human fibroblasts, seeded with melanoma cell lines (Fig. 2e and Supplementary Fig. 2c), and confirmed UV-HFF constructs presented fewer melanoma cells detaching from the tumour edge, singly advancing in the dermis (Sk-mel-28 $p = 0.04$; Fig. 2f). UV-HFF constructs replicated the cardinal features of UVR damage[34–36], with significantly reduced collagen levels compared to HFF constructs ($p < 0.0001$, Supplementary Fig. 2d, e). In addition, UV-HFF constructs presented reduced fibronectin (Supplementary Fig. 2f), and no difference in elastin expression compared to HFF constructs (Supplementary Fig. 2g). These data indicate that melanoma invasion is optimal within a range of collagen concentrations. Critically, lower collagen concentrations limit melanoma invasion.

Since UVR compromises collagen integrity indirectly by damaging fibroblasts (Fig. 1), we exposed melanoma spheroids, embedded in equal concentrations of collagen matrices, to increasing concentrations of the enzyme collagenase I to mimic the effects of UVR exposure (Supplementary Fig. 2h). We found that melanoma invasion and single cell invasion significantly decreased in matrices exposed to higher doses of collagenase I (5 μg/ml, $p = 0.0005$, 10 μg/ml, $p < 0.0001$; Fig. 2g–j and Supplementary Fig. 2i). These data show that collagen quantity and degraded collagen limit melanoma invasion.

To explore if adult fibroblasts regulate collagen degradation and melanoma invasion, we harvested adult dermal fibroblasts

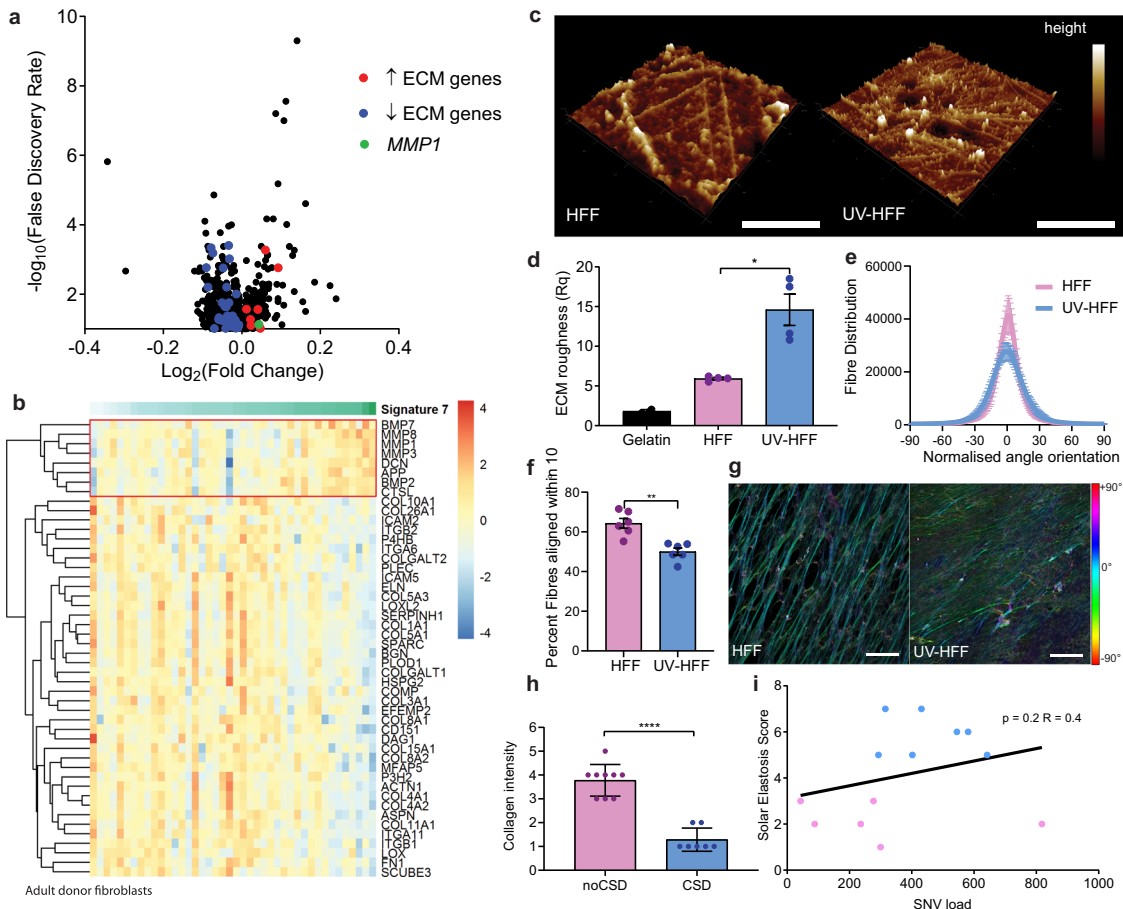

**Fig. 1 UVR-driven mutations in adult donor dermal fibroblasts correlate with ECM degradation and collagenase expression. a** Volcano plot gene expression and **b** extracellular matrix (ECM) heatmap by COSMIC signature 7 mutations in skin fibroblasts[23]. Colour scale: red upregulated, blue downregulated genes, sample clustered by signature 7 mutations: colour scale 0 (white) to 112 (green), red box: genes upregulated with signature 7 mutations. **c** Atomic force microscopy (scale: 5 μm, height colour scale −115 (dark brown), 115 (white)), and **d** matrix roughness (Rq) of in vitro ECM of HFF and UV-HFF, two-sided Mann–Whitney $U$ *$p = 0.0286$, data represents two measurements from two biologically independent cell lines per condition. **e** Quantification of fibre alignment distribution in human foreskin fibroblasts HFF and UV-HFF-derived ECM by fibronectin immunofluorescence. **f** Fraction of fibres within 10° of mode orientation, two-sided Mann–Whitney $U$ **$p = 0.0022$, data represents two biologically independent cell lines three fields of view per condition. **g** Immunofluorescence of fibronectin fibres in decellularised HFF and UV-HFF-derived ECM, colour coded for orientation of fibre, cyan represents mode, red ±90°, scale bar: 25 μm. **h** Masson's trichrome collagen stain of sun-protected ($n = 9$, noCSD) and sun-damaged ($n = 7$, CSD) dermis, two-sided Mann–Whitney $U$ ****$p < 0.0001$. **i** Correlation between SNV load and solar elastosis in fibroblasts, blue: CSD, pink: noCSD, two-sided Spearman correlation $R = 0.40$, $p = 0.2$, $n = 13$ biologically independent samples. Error bars: standard error of the mean (bar).

from different anatomic sites from tumour-adjacent normal skin of eight patients and established cells lines (Supplementary Table 3, age median 69, range 34–77). We confirmed the patient dermal fibroblasts express and secrete varying levels of MMP1 (Supplementary Fig. 2j, k), and embedded melanoma spheroids in matrices of collagen mixed with the donor fibroblast secretome. We found melanoma spheroids were less invasive in matrices containing patient fibroblast secretomes with higher amounts of MMP1 (Fig. 2k). Importantly, *COL1A1* expression correlated strongly with melanoma invasion, (Sk-mel-28 $p = 0.02$, $R = 0.32$, Fig. 2l). Finally, we confirmed that human fibroblasts, and not melanoma cells, are the main source of *MMP1*; and the melanoma cell lines do not express *COL1A1* or *COL1A2* (ref. [37], Supplementary Fig. 2l). Taken together, these data demonstrate human adult fibroblasts modulate collagen biology and melanoma invasion.

**Inhibition of MMP1 restores melanoma invasion**. We studied if adult human fibroblasts from distinct anatomic sites affect the

ECM and melanoma invasion differentially, and established fibroblast lines from chronic sun-damaged (CSD) and sun-protected, or non-sun damaged (noCSD) tumour-adjacent skin[32] of two patients (age CSD = 77, age noCSD = 46). Donor fibroblasts were exposed to low doses of UVB ($8 \times 100$ J/m²) to generate isogenic pairs of noCSD and noCSD-UV fibroblasts, CSD and CSD-UV fibroblasts and allowed 14 days recovery (Supplementary Fig. 2m). We confirmed noCSD-UV fibroblasts robustly increased *MMP1* expression (fold change = 2.03, $p = 0.002$) and secretion (median MMP1: noCSD = 6858 pg/ml, noCSD-UV = 13,292 pg/ml, fold change = 1.98, $p = 0.03$) compared to noCSD donor fibroblasts (Fig. 3a, b); and CSD-UV fibroblasts weakly upregulated *MMP1* expression (fold change = 1.46, $p = 0.39$), and secretion (median MMP1: CSD = 9066 pg/ml, CSD-UV = 13051 pg/ml, fold change = 1.49, $p = 0.03$). Importantly, noCSD-UV and CSD-UV fibroblasts did not increase *COL1A1* expression (Supplementary Fig. 2n). We then compared the effect of the fibroblast secretomes on melanoma spheroid invasion in the presence or absence of the MMP inhibitor Batimastat, which directly blocks the activity of MMPs.

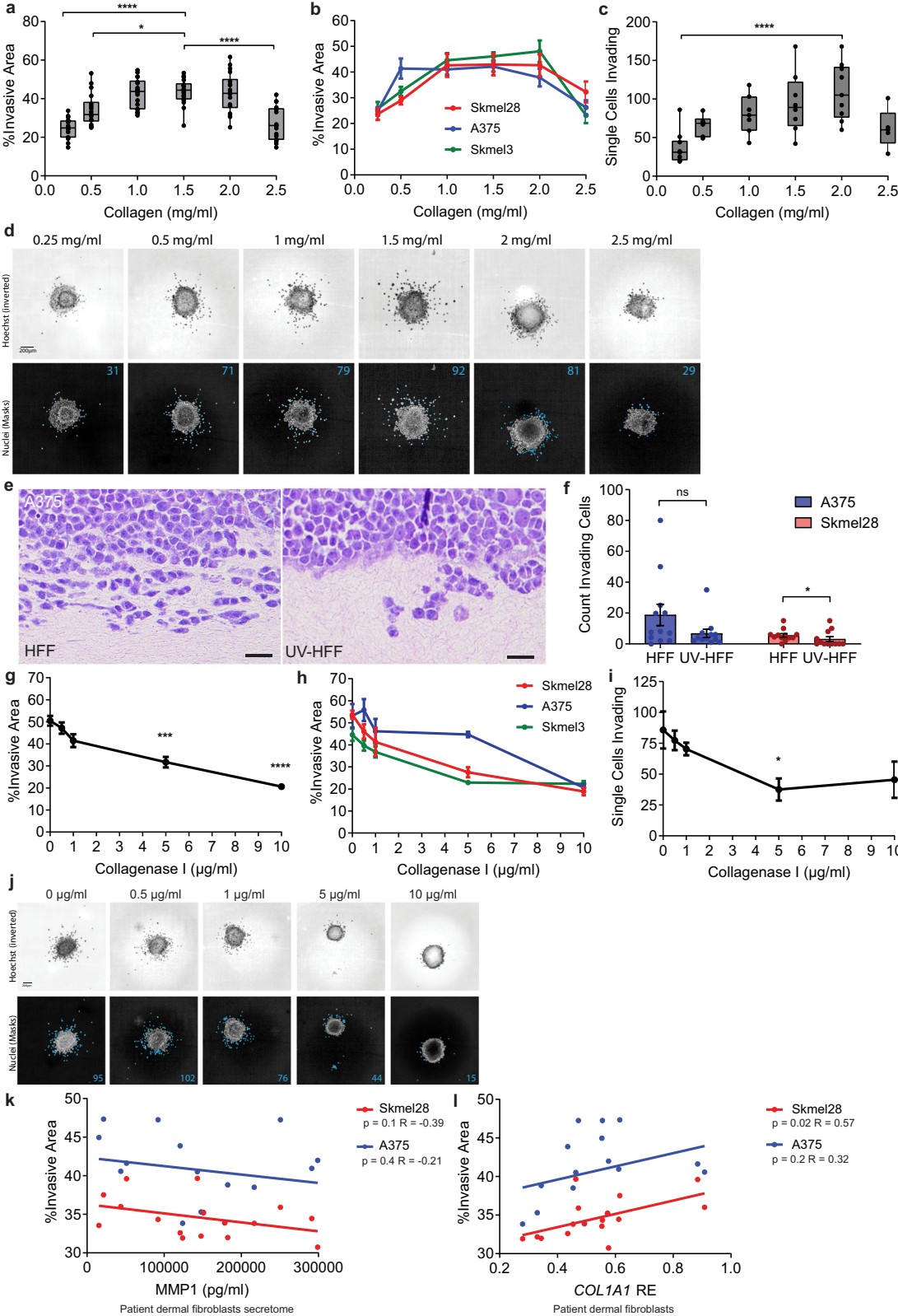

Consistent with a higher MMP1 expression, the noCSD-UV secretome significantly decreased melanoma invasion compared to the noCSD secretome ($p = 0.001$), and importantly, Batimastat restored melanoma cell invasion ($p = 0.01$; Fig. 3c). Intriguingly, UVR damage or Batimastat treatment of the CSD fibroblast model (CSD-UV) only slightly modulated melanoma invasion (Fig. 3d), possibly indicating the effect of UVR damage to fibroblasts and MMP1 expression is capped, and higher doses of MMP inhibition are required in highly expressing MMP1, CSD-UV fibroblasts.

**Fig. 2 Low collagen quantity and integrity decrease melanoma cell invasion. a** Mean and individual **b** melanoma spheroid invasion (Kruskal–Wallis with Dunn's multiple comparison tests *$p = 0.0317$, ****$p < 0.0001$, $n = 8$ replicate spheroids for three cell lines across two-independent experiments), and **c** single cell invasion (Skmel28) (Kruskal–Wallis with Dunn's multiple comparison tests ****$p < 0.0001$, $n = 9$ independent measurements over two experiments) in different collagen concentrations. **d** Representative images of spheroid and melanoma single cell invasion, top: Hoechst (inverted), bottom: invading cells (blue). **e** A375 melanoma invasion H&E (scale bar: 10 μm) and **f** single cell invasion in organotypic dermal collagen HFF and UV-HFF constructs (two-sided Mann–Whitney U *$p = 0.0485$, ns not significant, data represents 12 fields of view across two-independent experiments). **g** Mean and individual **h** melanoma invasion (Kruskal–Wallis with Dunn's multiple comparison tests ***$p < 0.001$, ****$p < 0.0001$, $n = 8$ replicate spheroids for three cell lines across two-independent experiments), and **i** melanoma single cell invasion (Kruskal–Wallis with Dunn's multiple comparison tests *$p = 0.0426$, $n = 7$ independent measurements over two experiments) by collagenase I concentration. **j** Representative images of spheroid and single cell invasion, top: Hoechst (inverted), bottom: invading cells (blue). **k** Melanoma spheroid invasion by adult patient fibroblast secretome, MMP1 levels (two-sided Pearson correlation, red: Sk-mel-28 $R = -0.39$ $p = 0.1$, blue: A375 $R = -0.21$ $p = 0.4$, data represent eight independent cell lines measured in duplicate) and **l** by *COL1A1* relative expression (RE) in fibroblasts (two-sided Pearson correlation, red: Sk-mel-28 $R = 0.57$ $p = 0.02$, blue: A375 $R = 0.32$ $p = 0.2$, data represent eight independent cell lines measured in duplicate). Error bars: standard error of the mean (bar).

Since MMP1 specifically cleaves COL1A1 (ref. [26]), we generated isogenic shCtrl-HFF, shCtrl-UV-HFF, sh*MMP1*-HFF and sh*MMP1*-UV-HFF lines from HFF to compare collagen degradation in the absence of acute UVR exposure (Supplementary Fig. 3b, c). In keeping with a higher expression of MMP1, shCtrl-UV-HFF fibroblasts degraded more collagen, while sh*MMP1*-UV-HFF did not increase collagen degradation compared to sh*MMP1*-HFF fibroblasts (Fig. 3e, f). The specific role of MMP1 was validated with an additional knockdown with an siRNA targeting *MMP1* (Supplementary Fig. 3d–f). In addition, we found shRNA targeting *MMP2* did not modify collagen degradation (Supplementary Fig. 3b, g, h); and knockdown of MMP1 restored the alignment of fibres in UV-HFF matrices (Fig. 3g, h and Supplementary Fig. 3i). Furthermore, organotypic invasion assays with matrices generated with shCtrl-HFF, shCtrl-UV-HFF, sh*MMP1*-HFF or sh*MMP1*-UV-HFF fibroblasts, showed melanoma invasion was decreased in the shCtrl-UV-HFF constructs ($p = 0.03$), and not in sh*MMP1*-HFF or sh*MMP1*-UV-HFF fibroblast matrices (Fig. 3i, j). Knockout of MMP1 restored collagen and fibronectin levels in UV-HFF constructs to similar levels as HFF (Supplementary Fig. 3j, k, l and see also Supplementary Fig. 2c, d, f). Altogether, these data demonstrate fibroblast-secreted MMP1 degrades collagen, limiting melanoma invasion.

**Collagen degradation decreases primary melanoma invasion and improves survival.** If the amount and integrity of collagen restricts single cell invasion, patients with primary cutaneous melanoma invading in a less collagenous dermis should live longer than patients with more dermal collagen. Compared to young fibroblasts and dermis, aged and UVR-protected fibroblasts in an aged ECM drive melanoma invasion and metastasis[38,39], so we restricted our study to three international cohorts of older primary cutaneous melanoma patients (Supplementary Table 4). We determined the proportion of melanoma cells invading in the ECM at the invasive front (IF), the amount of collagen and the degree of ECM degradation (solar elastosis) in tumour-adjacent dermis (Fig. 4a and Supplementary Fig. 4a). We found patient samples with more solar elastosis (less collagen in tumour-adjacent skin), had fewer invading cells at the IF (Fisher exact test, $p = 2.25 \times 10^{-5}$ Fig. 4b, Supplementary Fig. 4b and Supplementary Data 2). Critically, MSS was significantly improved in patients with less invasion in multivariate analyses (Fig. 4c, Supplementary Fig. 4c, d and Supplementary Table 5). Intriguingly, solar elastosis was not as powerfully associated with better outcome ($p = 0.9$, Fig. 4d, Supplementary Fig. 4e, f and Supplementary Table 5). To explain this difference, we hypothesised that collagen at the IF, rather than collagen degradation in the tumour-adjacent dermis, would be a better biomarker of survival. Further analysis confirmed that collagen at the IF strongly correlated to single cell invasion (Spearman R 0.5, $p < 0.0001$, Fisher Exact p $= 0.002$, Fig. 4e), MSS and progression-free survival (PFS; Fig. 4f, Supplementary Fig. 4g, and Supplementary Table 5). Furthermore, consistent with invasion data in Fig. 2j, TCGA primary cutaneous melanomas expressing low *COL1A1* showed improved survival (Fig. 4g and Supplementary Table 5). These data suggest that primary melanomas invading in collagen-poor matrices require new collagen synthesis in order to invade successfully. To confirm this, we studied collagen at the IF specifically, in short-term and long-term survivors. We confirmed melanomas arising at CSD sites with less collagen in the tumour-adjacent dermis and shorter survival (MSS <5 years), increased collagen deposition at the IF and tumour cell invasion (Fig. 4h, i and Supplementary 5).

To further explore the association between collagen and survival, we investigated if melanoma-associated fibroblasts (MAFs) in primary melanomas increase collagen to sustain invasion. For this, we extracted the gene expression signature from single cell RNAseq[37] and confirmed *COL1A* is expressed by MAFs (Supplementary Fig. 4h). We then tested if increased expression of MAFs in primary melanomas correlates with outcome and were able to demonstrate that a higher expression of MAF genes is associated with poor survival (Fig. 4j). Furthermore, we show that the expression of collagen genes specifically within the MAF expression signature is what impacts survival, as the MAF signature is not significantly prognostic in the absence of collagen genes (Supplementary Fig. 4i and Supplementary Table 5).

Solid cancers synthesise high amounts of ECM proteins and COL1A1; and ECM remodelling promotes primary tumour progression and metastasis[40,41]. Therefore, we designed a tumour-agnostic approach to test the potential of *COL1A1* expression as a biomarker for primary pan-cancer survival. This revealed young and aged patients with primary cancers expressing high levels of *COL1A1* are at greater risk of death and have shorter PFS (Fig. 4k, l and Supplementary Fig. 4j–l).

## Discussion

Multiple in vivo studies confirm UVR cooperates with oncogenic mutations to increase the incidence and penetrance of disease[5,42]. However, whether UVR exposure affects the odds of survival has not been comprehensively investigated, and there are contradictory studies finding sun exposure inferred by anatomic site[8], history of sunburn[9,18] or the presence of UVR-induced dermal degradation[6,10] can affect outcome. The majority of melanoma deaths affect the elderly[13], and age is strongly associated with accumulated sun exposure[7]. We investigated if pre-existing UVR damage affects melanoma

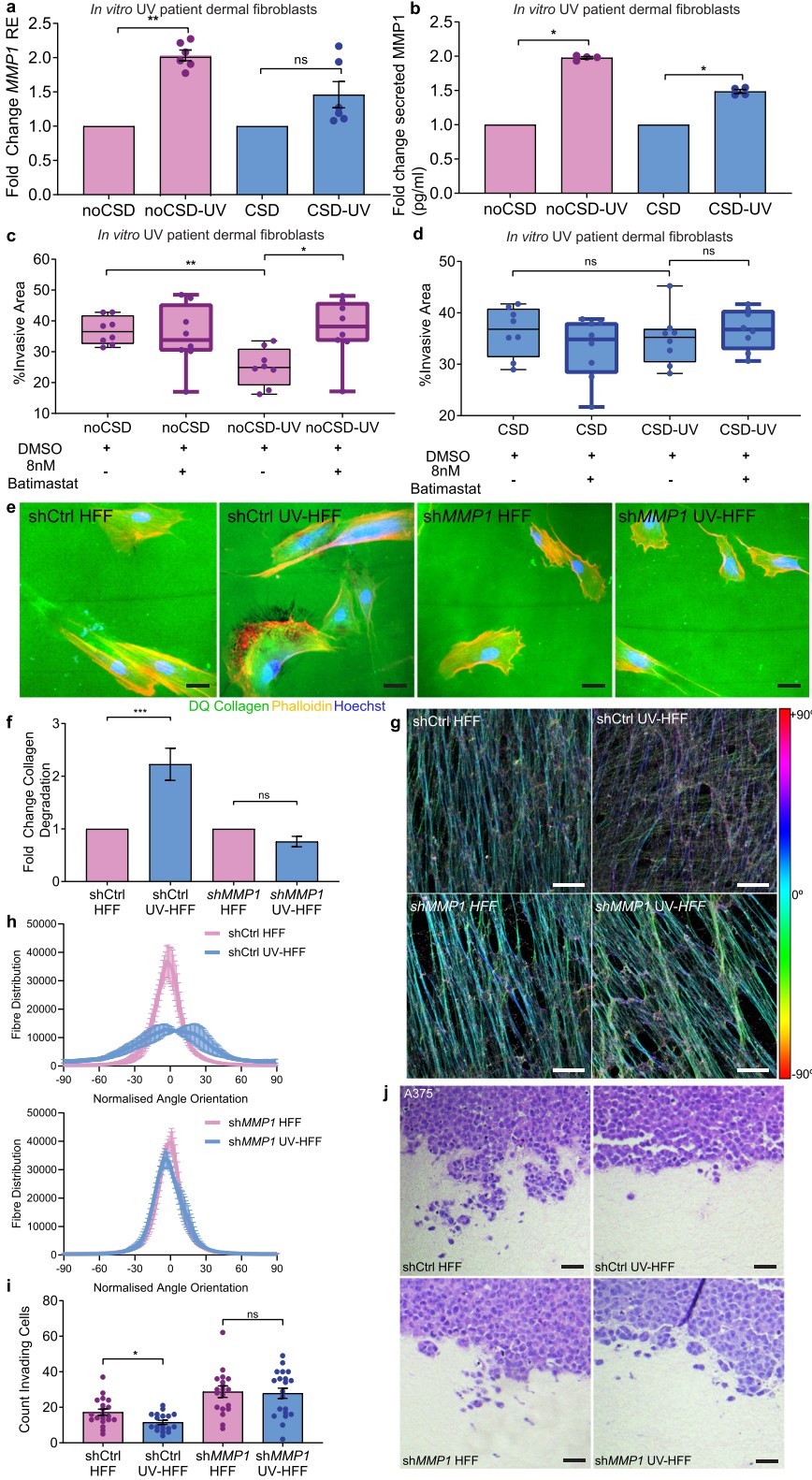

survival, and found low collagen quantity and integrity limit melanoma invasion. UVR damage to fibroblasts degrades collagen and the ECM, delaying melanoma progression. We confirmed our in vitro results, showing that in aged primary cutaneous melanomas, single tumour cells invading the dermis and collagen at the IF robustly predict poor survival. Paradoxically, this study finds UVR damage to the dermis destroys collagen, limiting invasion and improving outcome, unless tumours increase the production of collagen at the IF, providing the structural support for melanoma invasion. Together with recent work showing UVR-protected aged fibroblasts[38] and ECM[39] drive melanoma metastasis, this study strongly implicates the physical composition and structure of the aged tumour microenvironment, as key to primary melanoma

**Fig. 3 Inhibition of MMP1 restores melanoma invasion. a** Fold change in *MMP1* relative expression (RE) in chronically UVR-treated adult patient fibroblasts compared to untreated isogenic cell lines, noCSD no chronic sun damage, pink, CSD chronic sun damage, blue (two-sided Mann–Whitney *U* **p = 0.0022, ns not significant, data represent two samples per condition quantified in triplicate). **b** Fold change in secreted MMP1 in chronically UV-treated adult fibroblasts compared to untreated isogenic cell lines, noCSD no chronic sun damage, pink, CSD chronic sun damage, blue, (two-sided Mann–Whitney *U* *p = 0.0286, data represent two samples per condition quantified in triplicate). **c** Melanoma spheroid invasion in noCSD and isogenic noCSD-UV fibroblast secretomes in the presence of Batimastat or DMSO vehicle, (two-sided Mann–Whitney *U* **p = 0.0011, *p = 0.0104, data represents eight replicate spheroids measured across two-independent experiments per condition). **d** Melanoma spheroid invasion in CSD and isogenic CSD-UV fibroblast secretomes in the presence of 8 nM Batimastat or DMSO vehicle, (two-sided Mann–Whitney *U*, ns not significant). Data represents eight replicate spheroids measured across two-independent experiments per condition, box plots represent 25th to 75th percentiles with median, whiskers represent minimum and maximum values. **e** Representative images of collagen degradation in shCtrl-HFF, shCtrl-UV-HFF, sh*MMP1*-HFF and sh*MMP1*-UV-HFF fibroblasts. Green: intact DQ collagen; red: phalloidin; blue: Hoechst. Size bars: 20 μm. **f** Fold change in collagen degradation of shCtrl-HFF and sh*MMP1*-HFF (pink), and their isogenic chronic UVR cell lines shCtrl-UV-HFF and sh*MMP1*-UV-HFF (blue), (two-sided Mann–Whitney *U* ***p = 0.0007, ns not significant, n = 149 scores across two biologically independent cell lines per condition). **g** Immunofluorescence of fibronectin fibres in decellularised shCtrl-HFF, shCtrl-UV-HFF, sh*MMP1*-HFF and sh*MMP1*-UV-HFF-derived ECM, colour coded for orientation of fibre, cyan represents mode, red ±90°, scale bar: 25 μm. **h** Quantification of fibre alignment distribution in shCtrl-HFF, shCtrl-UV-HFF, sh*MMP1*-HFF and sh*MMP1*-UV-HFF-derived ECM, data represents z-stacks of three fields of view per sample. **i** Quantification of invading melanoma cells into organotypic dermal collagen constructs made with shCtrl-HFF and sh*MMP1*-HFF (pink), and isogenic chronic UVR cell lines shCtrl-UV-HFF and sh*MMP1*-UV-HFF (blue), (two-sided Mann–Whitney *U* *p = 0.0272, ns not significant, data represents >8 fields of view for two-independent experiments in biologically independent cell lines), scoring was performed on duplicate constructs counting in at least five fields of view per cell line. **j** Representative images of A375 invasion into organotypic constructs stained with H&E (scale bar: 10 μm). Error bars represent standard error of the mean (bar).

progression. We therefore infer from these joint studies that excessive old age mortality particularly affects patients with tumours arising at anatomic sites with preserved dermal collagen, or sun-protected skin. In contrast, UVR damage modifies the dermis and decreases collagen content as we age. As melanomas arising in collagen-poor, sun-damaged skin require collagen to invade, we show that collagen deposition at the invasive edge of the tumour is an independent, robust biomarker of survival. Conveniently, the deposition of collagen at the IF can be scored simply from haematoxylin and eosin stains, making this an ideal biomarker.

Melanomas with more UVR damage accumulate more mutations and neoantigens, possibly eliciting stronger immune responses[43,44]. However, we show a proportion of UVR melanomas have a better prognosis due to collagen degradation, independently of the mutation burden, tumour and stromal cell immunogenicity, which should be considered when evaluating responses to adjuvant immunotherapy. One possibility is to prioritise adjuvant care according to single cell invasion, collagen at the IF and risk of death. Supporting this rationale, recent evidence shows collagen density modifies the immune milieu of breast cancers, limiting T-cell responses[45].

The accumulation of a collagenous ECM, increasing stiffness, leads to poor prognosis and lack of response to therapies in other cancers[40]. Alterations of the ECM dynamics are a hallmark of cancer, able to deregulate cancer and stromal cells[46,47], promote cell transformation and the pro-metastatic niche, which becomes rich in vasculature and tumour-promoting inflammation[48]. Clinical trials with drugs inhibiting MMPs and limiting ECM remodelling have yielded negative and sometimes deleterious results. One possible explanation for this failure, based on our study in melanoma, is that inhibition of collagen degradation may support tumour invasion. Collagen can drive cancer cell de-differentiation[19,49–51] and is often found in areas of active epithelial cancer invasion, facilitating migration[50,52].

Ageing is associated with less collagen deposition, more degradation and higher overall cancer incidence and mortality in multiple tissues, including skin. One possibility is the collagen decrease in aged tissue, which could lead to less aggressive cancer, is offset by a decrease in ECM structural fitness, collagen and matrix organisation, and pro-tumourigenic signalling with age[41].

This work suggests new collagen synthesis by the TME is a critical regulator of aged primary melanoma progression, a feature that could drive poor outcome in other aged cancers. Critically, our data shows collagen expression is associated in multiple solid epithelial and non-epithelial primary tumours with shorter PFS in all ages, possibly due to direct modulation of invasion.

## Methods

**Cell lines and patient fibroblasts**. HFF were purchased from ATCC (ATCC SCRC-1041). Three melanoma cell lines, Sk-mel-28 (ATCC HTB-72), Sk-mel-3 (ATCC HTB-69) and A375 (ATCC CRL-1619) were purchased from ATCC. All cell lines were cultured in DMEM (Gibco, 41966-029) supplemented with 10% FCS (Sigma Aldrich, F7524), 1× Glutamax (Gibco, 35050061), 100 U/ml penicillin and streptomycin (Gibco, 15140122) and 1 mM sodium pyruvate (Gibco, 11360070). Cells lines were cultured at 37 °C in 5% $CO_2$ with medium replaced as required. Cell lines were tested every fortnight for mycoplasma using LookOut Mycoplasma PCR Detection Kit (Sigma Aldrich, MP0035). Cell line identity was confirmed using STR profiling.

**Patient fibroblasts**. A prospective cohort of patient fibroblast cultures was established from redundant skin acquired during surgical resection of the wide local excision of healthy skin from melanoma patients treated at the tertiary referral cancer Christie hospital. Ethical approval to establish cell lines was granted by the local Biobank committee (17_AMVI_01), which required signed informed consent from all participants. The hypodermis of whole skin samples was removed scraping with a scalpel, and the residual specimen was incubated overnight in Dispase (Gibco, 17105-041) at 4 °C to separate the epidermis and dermis. The dermis was digested in collagenase I (Gibco, 17018029) in DMEM (without FCS) at 37 °C for 6 h, and then filtered through 70 μm filter to remove the residual debris. Dermal cells were spun at $300 \times g$ and resuspended in DMEM 20% FCS, cultured in DMEM 20% FCS until they became confluent and stained for vimentin (Abcam, ab92547). The level of solar elastosis of the redundant skin collected was scored using previously well-established methods[32].

**Whole-exome sequencing of patient fibroblasts**. Whole-exome sequencing of fibroblasts was performed by Novogene (Novogene (UK) Company Ltd.). Exome capture was performed with the SureSelect Human All Exon v6 kit (Agilent) and sequenced on the Illumina HiSeq platform. Sequencing reads were trimmed using Trimmomatic[53], aligned to the hg38 reference genome using BWA[54], and duplicate reads were marked using Picard Tools (http://broadinstitute.github.io/picard). Somatic mutations were called using the Varscan 2 pipeline[55]. Identified somatic variants were annotated using Variant Effect Predictor[56] and variants present in dbSNP (but not in the COSMIC database) were excluded.

**Lentiviral shRNA transfection**. Knockdown of *MMP1* expression in HFF cells was performed using shRNA Lentiviral Particles (Santa Cruz Biotechnology) and siGENOME siRNA (Horizon Discovery). For *MMP1* shRNA knockdown MMP1 shRNA (h) lentiviral particles (sc-41552-V) were used alongside a scramble control, control shRNA lentiviral particles A (sc-108080) and copGFP

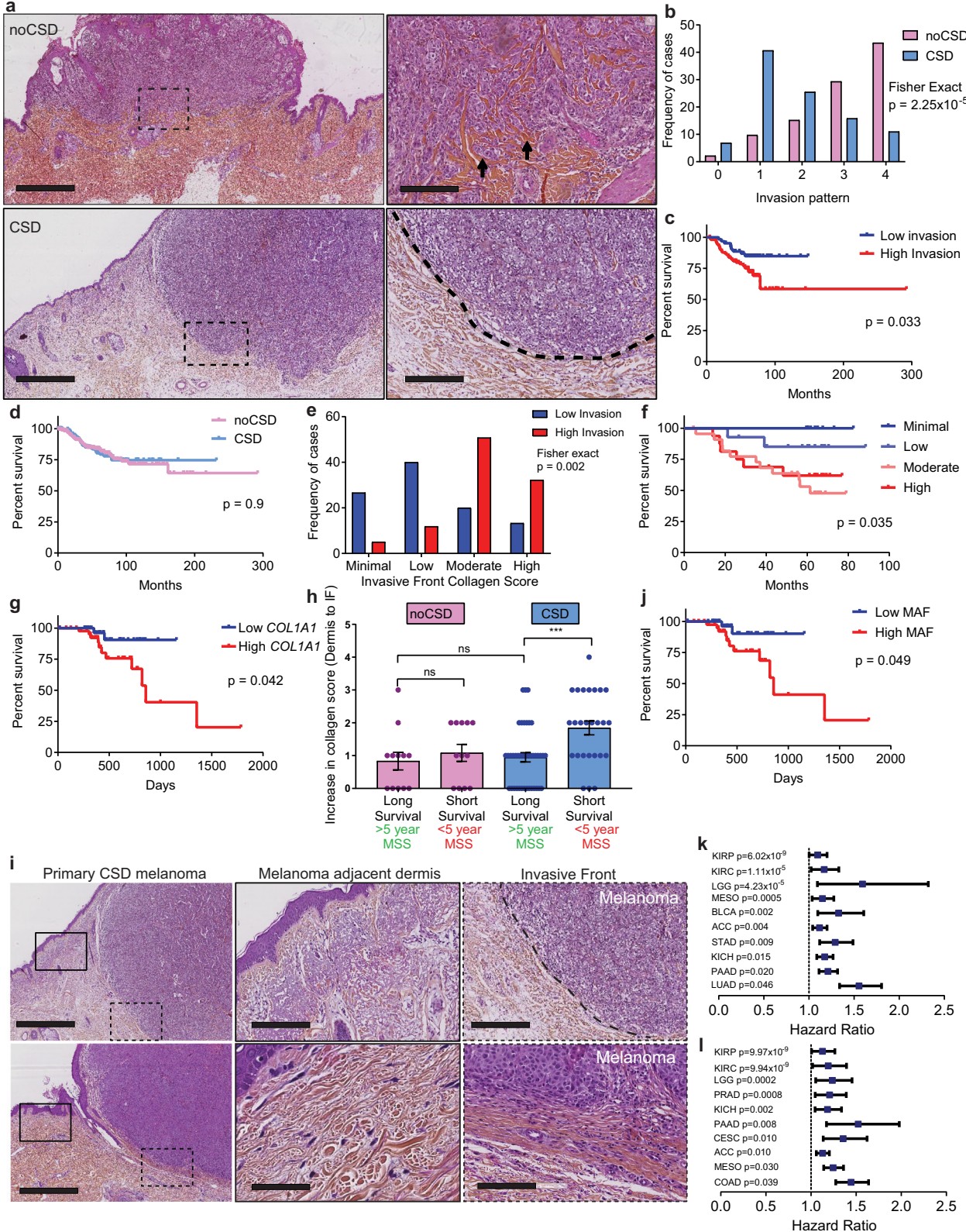

control lentiviral particles (sc-108084) were used to measure transduction efficiency. A total of $5 \times 10^4$ cells were cultured in cell culture media with 5 ug/ml polybrene (Santa Cruz Biotechnology, sc-134220). Lentiviral particles were added to cells and incubated overnight. Media containing lentiviral particles and polybrene was removed and incubated in DMEM overnight before preforming selection of transfected cells using increasing concentration of puromycin over 72 h. Once cells were stably growing in puromycin, cells were cultured as normal

in DMEM. For *MMP1* siRNA siGENOME human MMP1 siRNA (D-005951-02-0002) and non-targeting siRNA #4 (D-001210-04-05) were used with DharmaFECT 1 transfection reagent (Horizon Discovery, T-2001-02), according to manufactures protocols, with a final siRNA concentration of 25 nM. Knockdown of *MMP2* was performed with MMP2 shRNA (h) lentiviral particles (sc-29398-V) as above. Knockdown of all gene expression was validated by qPCR and western blot.

**Fig. 4 Low collagen correlates with low invasion and improved outcome in aged melanoma patients. a** H&E top panel: primary cutaneous melanoma and inset (box) with single cell invasion (arrows) in a collagen-rich dermis with no chronic sun damage (noCSD), collagen: red (scale bars left: 4000 μm, right: 400 μm). Lower panel: melanoma in a collagen-poor dermis with chronic sun damage (CSD) and inset (box), no single cell invasion; collagen: red (scale bars left: 3000 μm, right: 300 μm). **b** Histogram displaying melanoma invasion at the invasive front (IF) in noCSD and CSD melanomas (B and C cohorts, $n = 170$, two-sided Fisher exact test, $p = 2.25 \times 10^{-5}$). **c** Kaplan–Meier of melanoma-specific survival (MSS) in prominent (high, red) and minimal (low, blue) melanoma invasion at the IF (two-sided log-rank test, B and C cohorts, $n = 167$). **d** Kaplan–Meier of MSS in melanoma invading in CSD (blue) and noCSD (pink) dermis (two-sided log-rank test, B and C cohorts, $n = 331$). **e** Histogram displaying collagen quantity at the IF in highly invasive (red) and minimally invasive (blue) melanoma (C cohort, $n = 89$, two-sided Fisher exact test, $p = 0.002$). **f** Kaplan–Meier of MSS by collagen quantity at the IF (two-sided log-rank test, C cohort, $n = 62$). **g** Kaplan–Meier of MSS by *COL1A1* expression in aged (>54) primary cutaneous melanoma (two-sided log-rank test, TCGA cohort, $n = 80$). **h** Fold increase in collagen deposition at the IF of noCSD and CSD melanomas by MSS (two-sided Mann–Whitney $U$ ***$p = 0.0009$ C cohort $n = 90$). Error bars represent standard error of the mean (bar). **i** H&E top panel: left CSD melanoma, middle: from box inset: tumour-adjacent dermis; right: from dashed box inset: IF (dashed line, scale bars: 2000, 300, 300 μm). Bottom: left CSD melanoma, middle: from box inset: tumour-adjacent dermis; right: from dashed line box inset: IF between dashed lines, arrows: melanoma invasion, (scale bars: 2000, 70, 200 μm, $n = 90$). **j** Kaplan–Meier of MSS by melanoma-associated fibroblast (MAF) signature score in aged (>54) primary cutaneous melanoma cohort (two-sided log-rank test, TCGA cohort, $n = 80$). **k** Hazard ratio (centre) and 95% CI (bars) for OS, and PFS (**l**) in two-sided univariate Cox regression of *COL1A1* expression by cancer type in PANCAN TCGA, $p$ values unadjusted. (ACC $n = 79$, adrenocortical carcinoma, BLCA $n = 407$, bladder urothelial carcinoma, CESC $n = 304$, cervical and endocervical cancers, COAD $n = 448$, colon adenocarcinoma, KICH $n = 65$, kidney chromophobe, KIRC $n = 533$, kidney renal clear cell carcinoma, KIRP $n = 289$, kidney renal papillary cell carcinoma, LGG $n = 514$, brain lower grade glioma, LUAD $n = 506$, lung adenocarcinoma, MESO $n = 86$, mesothelioma, PAAD $n = 178$, pancreatic adenocarcinoma PRAD $n = 497$, prostate adenocarcinoma, STAD $n = 409$, stomach adenocarcinoma). Risk tables for all Kaplan–Meier analyses in Supplementary Data 2.

**UV treatment and CSD model.** Cell lines were treated with UVB using a Bio-Sun UV irradiation system (Vilber Loumat). For chronic treatment the dose 100 J/m$^2$ was used as it represents a physiologically relevant dose of UVB that would penetrate the dermis between 1 and 5 minimal erythema dose[57]. To create isogenic in vitro chronic UV-damaged UV-HFF and sh*MMP1*-UV-HFF, noCSD-UV, CSD-UV cell lines, $1 \times 10^6$ HFF, sh*MMP1*-HFF or patient dermal fibroblasts were cultured in 100 mm dishes in phenol-free DMEM 1% FCS. All fibroblasts were treated every 24 h with 100 J/m$^2$ UVB for eight consecutive days. Following the UV treatments, the medium was changed to DMEM 10% FCS and cultured for 1 week. Isogenic untreated control cell lines (HFF, sh*MMP1*-HFF, noCSD and CSD) were cultured in identical conditions without the UVB treatments. Each HFF condition was created in biological duplicates.

**Secretome collection.** To collect secretomes $1 \times 10^6$ cells were plated in a 100 mm dish and cultured for 72 h in DMEM without FBS to limit cell proliferation. Secretomes were collected in duplicate, aliquoted and stored at −80 °C until used.

**Batimastat treatment.** MMP inhibitor Batimastat (Sigma Aldrich, SML0041) was resuspended in DMSO at 15 mg/ml and a stock was diluted to 1 mM. Batimastat was added to fibroblast secretomes to a final concentration of 8 nM in secretome volume and control secretomes had equal volume of DMSO added as a vehicle control.

**RNA-sequencing data analysis.** RNA-seq data from ENA project PRJEB13731; also at https://www.ebi.ac.uk/arrayexpress/experiments/E-MTAB-4652/ were downloaded. Data are single-end RNA-seq from short-term cultivated fibroblasts sequenced on an Illumina HiSeq 2000 sequencer. Two samples had been obtained from each individual from different locations (B = buttock, not UV exposed; S = shoulder, UV exposed). Sequencing reads were trimmed[53] and aligned to the human reference genome (GRCh37) using STAR[58]. Production of analysis-ready reads was conducted according to the Broad Institute Best Practices pipeline (https://gatkforums.broadinstitute.org/gatk/discussion/3892/the-gatk-best-practices-for-variant-calling-on-rnaseq-in-full-detail) using GATK v3.2 (http://www.broadinstitute.org/gatk). Somatic single nucleotide variants (SNV) in each patient matched sample in regions annotated as protein-coding only (based on Ensembl Homo_sapiens.GRCh37.87) were identified with the mutation calling algorithm MuTect v1, using the other sample as the comparator[59]. Identified somatic variants were annotated using Variant Effect Predictor and common variants were excluded.

COSMIC mutational signatures v2 (https://cancer.sanger.ac.uk/cosmic/signatures_v2) were identified using the MutationalPatterns[60] package (version 1.8.0) in R (version 3.5.1, RStudio v1.2.5001, RStudio Inc). Differential expression analysis was performed using the DESeq2 package (version 1.22.2, ref. [61]) in R (version 3.5.1). Reads counts of genes were filtered for genes expressed in fibroblasts by removing any gene with <100 counts across all samples. For pathway enrichment analysis genes that were significantly differentially expressed in by signature 7 mutation count (false discovery rates (FDR) $p$ value < 0.1) were compared against the Reactome Database[62] using the Molecular Signatures Database v7.0 (Msigdb, Broad Institute, https://www.gsea-msigdb.org/gsea/msigdb/index.jsp).

**Fibroblast extracellular matrix production.** Following ECM construction[63] cell culture dishes were coated with 0.2% sterile gelatin (Sigma Aldrich, G1393), fixed with 1% glutaraldehyde and quenched with 1 M glycine in PBS (pH 7). Fibroblasts were cultured on gelatin plates in normal 10% FCS DMEM containing 50 μg/ml ascorbic acid for 8 days. Cells were lysed with extraction buffer (20 mM NH$_4$OH and 0.5% Triton X-100 in PBS), and washed thoroughly with PBS containing calcium and magnesium. DNA was digested with 10 μg/ml DNase I (Roche, 04716728001) and washed. For mass spectrometry the ECM was collected with a lysis buffer (100 mM TrisHCl pH 7.5, 4% SDS and 100 mM DTT) and collected with scraper, sonicated and boiled at 95 °C, followed by centrifugation (16,000 × $g$, 15 min) and collection of the supernatant, stored at −80 °C until use.

**Immunofluorescence and ECM fibre analysis.** For immunofluorescence and ECM fibre analysis, fibroblasts derived matrices constructed as above on glass coverslips, after DNase digestion matrices, were fixed in 4% paraformaldehyde (PFA), blocked 5% BSA TBS-T (5% BSA in 1× Tris-buffered saline and 0.1% Tween 20) for 1 h, and stained for Fibronectin[64]. Primary antibody to fibronectin (1:200, F3648, Sigma Aldrich, 5% BSA TBS-T) and secondary goat anti-rabbit Alexa Fluor 488 (1:2000, Thermo Fisher) were incubated for 1 h at room temperature. The immunofluorescence imaging was performed using a Carl Zeiss LSM880 inverted confocal microscope with 63× NA 1.4 oil objectives lens controlled by ZEN black software. The images were acquired with 488 nm illumination laser line from an Argon laser (Lasos) and the emission spectrum range from 500 to 550 nm collected with a PMT detector (Zeiss). Z-series optical sections were collected with a step size of 0.5 micron driven by Piezo stage (Zeiss). Fibre orientation analysis was performed using ImageJ OrientationJ plugin[64,65]. Maximal projection of three individual z-stacks for each condition were analysed.

**Mass spectrometry sample preparation and analysis.** ECM protein lysates were separated on a 4–12% gradient NuPAGE Novex Bis-Tris gel (Life Technologies). Each sample was cut into three slices and in-gel digested with trypsin (Promega)[66,67]. Digested peptides were desalted by C18 StageTip[68], acetonitrile was removed by speed vacuum, and peptides were resuspended in 1% trifluoroacetic acid and 0.2% formic acid. Peptides were injected into an EASY-nLC (Thermo Fisher Scientific) coupled online to an Orbitrap Fusion Lumos mass spectrometer (Thermo Fisher Scientific), separated using a 20 cm fused silica emitter (New Objective) packed in house with reversed-phase Reprosil Pur Basic 1.9 μm (Dr Maisch GmbH) and eluted with a flow of 300 nl/min from 5 to 30% of buffer (80% ACN and 0.1% formic acid), in a 90 min linear gradient. MS raw data were acquired using the XCalibur software (Thermo Fisher Scientific). MS raw files were processed using MaxQuant software[69] (version 1.6.3.3) and searched against the human UniProt database (release 2016_07, 70,630 sequences), using the Andromeda search engine[70] with the following settings: the parent mass and fragment ions were searched with an initial mass deviation of 4.5 and 20 p.p.m., respectively. Carbamidomethyl (C) was added as a fixed modification and acetyl (N-term) and oxidation (M) as variable modifications. The minimum peptide length was set to seven amino acids and a maximum of two missed cleavages, and specificity for trypsin cleavage were required. The FDR at the protein and peptide level were set to 1%. The LFQ setting was enabled for protein quantification[71]. Razor and unique peptides were used for quantification. Perseus software[72] (version 1.6.2.2) was used for statistical analysis. Data were filtered to remove potential contaminants, reverse peptides that match a decoy database and peptides only identified in their modified

form. LFQ intensities were transformed by log$_2$. A two-sample $t$ test was used to determine significantly regulated proteins, with the permutation-based FDR ≤ 0.05 and $S0 = 0.1$ being considered significant.

**Atomic force microscopy**. For imaging purposes, the ECMs prepared as above were fixed with 2% PFA and stored with PBS containing 1% penicillin/streptomycin at 4 °C. A day prior to imaging, the dishes were washed with distilled water five times to wash off any salt and then air dried overnight. Samples were imaged by intermittent contact mode in air using a Bruker ScanAsyst 9.1. The probe was auto-tuned using Nanoscope software (version 1.4). Images were taken at $10 \times 10$ μm and $2 \times 2$ μm area at least two sites. Data were processed using Nanoscope analysis software 1.4 prior to image export. The roughness (Rq) values were determined using the software. Roughness is the root mean square average of the image and is calculated based on the height difference per pixel along the sample length. Rq is used to study the surface topography of various nanostructures[28,29]. Rq provides a quantitative measure of fibril organisation in dermis and could possibly suggest the integrity of matrix[30].

**Collagen degradation assay**. To quantify the degradation of collagen in different fibroblast cell lines a collagen degradation assay based on ref. [73] was used. DQ collagen, type I from bovine skin, fluorescein conjugate (Invitrogen, D12060) was used to coat the wells of a 96-well plate for 1 h at 37 °C and quenched with 20 mM glycine for 5 min. Next, fibroblasts were plated at a concentration of $2.5 \times 10^3$/well and incubated in normal culture conditions for 18 h. Cells were then fixed with 4% PFA for 20 min at room temperature, permeabilised with Triton 0.1% X-100 for 5 min at room temperature and stained with Alex Fluor 546 Phalloidin (Invitrogen, A22283) 1:2000 for 1 h and DAPI (Invitrogen, D3571) 1:2000 for 15 min. Wells were imaged with the Opera Phenix High Content Screening System (Perkin Elmer, Inc.), and collagen degradation was quantified by either measuring the area, where DQ collagen had been degraded and normalising to cell number (DAPI) or by scoring of images by two-independent assessors (one of them blinded), scoring the intensity of collagen degradation as low (1: peri-cytoplasmic ring of collagen degradation <1/4 of the cytoplasmic diameter); medium (2: peri-cytoplasmic ring of collagen degradation ~1/3 of the cytoplasmic diameter) or high (3: peri-cytoplasmic ring of collagen degradation ~1/2 of the cytoplasmic diameter; Supplementary Fig. 3a). To establish the ratio of degradation between shCtrl-HFF and shCtrl-UV-HFF, and sh*MMP1*-HFF and sh*MMP1*-UV-HFF cells, we generated a score ($H$) of collagen degradation for each condition:

$$H = \Sigma \left(\text{low intensity images} \times 1\right) + \left(\text{medium intensity images} \times 2\right) + \left(\text{high intensity images} \times 3\right) \div \text{total images}$$

We scored, 150 images for shCtrl-HFF, 147 images for shCtrl-UV-HFF, 150 images for sh*MMP1*-HFF and 146 images for sh*MMP1*-UV-HFF fibroblasts. The ratios were established as: ratio shCtrl-UV-HFF/shCtrl-HFF = $H$-score shCtrl-UV-HFF/$H$-score shCtrl-HFF, and ratio sh*MMP1*-UV-HFF/sh*MMP1*-HFF = $H$-score sh*MMP1*-UV-HFF/$H$-score sh*MMP1*-HFF.

**MMP1 ELISA**. MMP1 in the secretome of cell lines was quantified with a MMP1 Human ELISA Kit (Thermo Fisher Scientific, EHMMP1), according to manufacturer's protocol. Secretomes collected from cells were diluted 1:10 and standards and samples were measured in triplicate. Samples were incubated on plate overnight at 4 °C. Absorbance measured on a Spectra Max M5 plate reader (Molecular Devices).

**Quantitative PCR**. RNA was collected in duplicate from $1 \times 10^6$ cells lysed in TRIzol Reagent (Invitrogen, 15596018) after secretomes were collected. The aqueous phase of phenol–chloroform separation was collected and RNA extracted using RNeasy Mini Kit (Qiagen, 74104). Concentration was determined with the Qubit RNA HS Assay (Invitrogen, Q32852) and 500 ng RNA was reverse transcribed to cDNA using TaqMan Reverse Transcription Reagents (Thermo Fisher, N8080234), and diluted 1:20 in nuclease free water. Genes were quantified by qPCR using TaqMan Gene expression assays and Fast Mastermix on a QuantStudio 3 system. *GAPDH* (Hs02758991_g1) and *ACTB* (Hs01060665_g1) were used as housekeeping genes. *MMP1* (Hs00899658_m1), *COL1A1* (Hs00164004_m1) and *COL1A2* (Hs01028956_m1) were quantified and normalised to the geometric mean of both housekeeping genes and relative expression calculated using $2^{-\Delta ct}$.

**Western blots**. Intracellular protein was extracted from cells using the NucBuster Protein Extraction Kit (Merck, 77183) and quantified using Pierce BCA Protein Assay Kit (Thermo Fisher Scientific, 23225). A total of 40 μg of cytoplasmic protein was diluted in laemmli buffer (Bio-Rad, 1610747) with beta-mercaptoethanol, denatured at 95 °C for 5 min and loaded onto Mini-PROTEAN TGX Gels (Bio-Rad, 4568084). Samples were transferred to nitrocellulose membranes using the TransBlot Tubro system (Bio-Rad, 170–4270) and protein visualised with Ponceau Stain (G-Biosciences). Membranes were blocked in 5% BSA TBS-T (5% BSA in 1× Tris-buffered saline and 0.1% Tween 20) for 1 h and incubated with primary antibodies overnight in 5% BSA TBS-T (MMP1 1:1000 ab137332, MMP2 1:1000 D4M2N Cell Signalling, B-actin 1:10,000 ab8226, Abcam). Membranes were

washed with TBS-T and incubated in secondary antibodies (Dnk pAb to Rb IgG IRDye 680RD, ab216779, Goat pAb to Ms IgG IRDye 800CW, ab216772, Abcam) for 1 h at room temperature. Membranes were visualised using the Odyssey CLx system (Licor).

**Melanoma spheroid invasion assay**. Melanoma cell lines were cultured in U-bottom 96-well plates (Brand, 781900) at $1 \times 10^3$ cells per well, spun at $200 \times g$, and spheroids allowed to form over 72 h. Culture media was removed from the wells and 100 μl collagen (PureCol, Advanced BioMatrix, 5005-100 ML) added to the wells. Plates were briefly spun for 15 s at $200 \times g$ and incubated at 37 °C for 1 h to set collagen. For concentration gradient collagen was diluted to 0.25, 0.5, 1, 1.5, 2 or 2.5 mg/ml in phenol-free DMEM without FCS. For degraded collagen invasion collagen was diluted to 1.5 mg/ml in phenol-free DMEM containing collagenase I (Gibco) to final concentration in the collagen of 0.5, 1, 5 or 10 μg/ml. For invasion using fibroblast secretomes collagen was diluted to 1.5 mg/ml using secretome collected from various fibroblast cell lines instead of DMEM. Spheroids were allowed to invade over 72 h and light microscopy photographs were taken. Images were analysed to quantify the invasive area around the spheroid by creating a layered mask for the spheroid core and invasive area, and quantifying the invasive area as a percentage of the total size of the spheroid in ImageJ software (1.53c). For single cell invasion analysis, spheroids in the above conditions were imaged using an Opera Phenix (Perkin Elmer) with a 5× (0.16 NA) lens. Spheroids were stained with Hoechst 33342 for 1 h prior to imaging. Stacks of images were acquired through the full depth of the spheroids, and images were analysed using Columbus software (Perkin Elmer). A custom analysis pipeline was used to detect individually invading cells. In brief, a maximal intensity projection of the Hoechst stain was processed (Flatfield correction: Basic, Guassian blur: 2px) was used to determine the perimeter of the spheroid, and nuclei were detected in the remaining portion of the image. Experiments with varying levels of collagen matrices or collagenase were repeated, and confirmed by an independent laboratory, blinded for matrix composition.

**Organotypic 3D invasion models**. Melanoma invasion through fibroblast-modified collagen was assayed using a protocol adapted from Timpson[74]. Briefly, equal numbers of HFF, UV-HFF, sh*MMP1*-HFF and sh*MMP1*-UV-HFF fibroblasts were mixed with collagen I, rat tail (Corning, 354236) and cultured in 35 mm culture dishes. Collagen discs were allowed to contract until they fit in a 24-well plate. Cell suspensions of Sk-mel-28 and A375 at $4 \times 10^4$ cells/ml were plated on top of each collagen disc in duplicate for each fibroblast condition. Cells and collagen were cultured as normal for approximately 5 days. Collagen discs were then transferred to Falcon 3.0 μm high density PET membrane (Corning, 353092) in Falcon six-well Deep Well TC-treated Polystyrene Plates (Corning, 355467) to create an air/liquid interface to drive melanoma invasion into collagen. After 10 days constructs were fixed in 4% PFA, and embedded in paraffin and stained with H&E and stained for Fibronectin with FN1 antibody (F3648, Sigma Aldrich). For each construct, the number of cells invading into the collagen was counted in at least five different fields of view under light microscopy, by two-independent scorers. Leica SCN400 was used for whole slide imaging alongside ImageScope software v12.3 (Leica Microsystems).

**Second-harmonic generation imaging**. Second-harmonic generation imaging of collagen in 3D organotypic models was performed using a Leica SP8 upright confocal microscope with 25× NA 0.95 water objectives controlled by Leica LAS X software. The images were acquired with 880 nm illumination laser line from MaiTai Ti:Sapphire laser (Spectra Physics) and HyD-RLD detector installed 440/20 nm filter cube (Leica), also used 483/32 nm filter (Leica) collecting auto-fluorescence signals at the same time. Z-series optical sections were collected with a step size of 1 micron driven by SuperZ galvo stage (Leica). Collagen was quantified in ImageJ by measuring the mean signal intensity of $z$-stack sum projections in three equal sized areas across three fields of views for each collagen disc.

**Clinical samples**. Three international patient cohorts of primary cutaneous melanoma were used in this paper. The A cohort ($n = 31$), the B cohort ($n = 222$) and the C cohort ($n = 113$). (A: Salford Royal NHS Foundation Trust, UK, B: Instituto Valenciano de Oncología, Valencia, Spain, C: Aix-Marseille University Hospital, France). Comprehensive clinical outcome was available for the B and C cohorts; and was collected prospectively at both institutions. All clinical and pathological information assessed complied with all relevant ethical regulations for work with human participants in the UK, Spain and France: Salford cohort A: Local ethics and UK NHS REC regulation approval, IRAS 16/LO/2098 (16/SW/0323); no patient signed consent required; Spanish cohort B: Internal Review Board of the Instituto Valenciano Oncología in Valencia, and verbal informed consent was obtained from all patients who were alive at the time of the study; French cohort C: Internal Review Board of the Comite de Protection des Personnes Sud Médi-terranée and Aix-Marseile University Hospital approval, and signed informed consent was obtained. The analyses were performed by at least two observers (cohort A: observer A.V. and L.M.; cohort B: V.T., E.N. and A.V.; cohort C: A.V. and L.M.). Discrepancy in cohorts A and B were jointly reviewed and consensus

agreed. There was high kappa interobserver agreement in cohort C (>0.65), and all scores were done blinded for clinical outcome.

Comprehensive clinical outcome was available for the B and C cohorts; and was collected prospectively at both institutions. The correlation between solar elastosis and invasion of melanoma cells at the IF was done in the A cohort in patients with invasive primary melanoma, where a distinct vertical growth was determined in patients aged ≥55 at the time of diagnosis. The correlation and histological assessments of the B and C cohorts were done in primary cutaneous melanomas of patients aged ≥55 at the time of diagnosis with Breslow ≥1 mm. The clinical characteristics of the cohorts are described in Supplementary Table 4.

**Histological and clinical sample analysis**. The histological assessment of primary cutaneous melanomas of the A, B and C (20%) cohorts, from three international centres, was performed by at least two observers (cohort A: observer 1 and 2; cohort B: 1, 3 and 4; cohort C: 1 and 2). Discrepancy in cohorts A and B were jointly reviewed and consensus agreed. There was high interobserver agreement in cohort C (>0.65), and all scores were done blinded for clinical outcome. The survival analyses were performed by members of the team who did not score histological variables. We included all samples with sufficient material to assess the tumour body, IF and tumour-adjacent skin. Recurrent tumours were excluded. Non-primary melanomas were excluded. Solar elastosis was scored as described by Landi[32] et al., and cutoffs for low, moderate and high solar elastosis; or CSD noCSD established from the original Landi categories. Landi et al. established a scoring system for the degree of solar elastosis from absent to severe using an 11-point score, from 0 to 3+. To generate binary categories, cases are classified as bearing no chronic sun damage (noCSD), for scores between 0 and 2−, or CSD for scores 2 to 3+. Cutoffs for absent (range 0, 0+), low (range 1− to 1+), moderate (range 2− to 2+) and high (3− to 3+) were established from the same range[75]. We assessed the inter-reliability of the binary CSD classification between two scorers using the kappa statistic, which showed 0.75 concordance for the B cohort (weighted kappa = 0.75, 95% CI = 0.69–0.79).

The proportion of melanoma cell invasion at the IF in the dermis was scored in categories. We assessed the front of the melanoma component in the dermis that is in direct contact with the dermal matrix, and scored 0/1: no invasion/minimal invasion: <5% of cells in contact with the dermis are actively invading the matrix, detaching from the IF of the melanoma; 2: low invasion: 5–25% of melanoma cells at the IF detaching from the tumour body; 3: moderate invasion: 25–50% of the IF is actively detaching from the main VGP and entering deeper structures; 4: high invasion: the majority of cells at the IF are independently interacting with the matrix, detached from the body of the tumour. Binary categories were then generated with low invasion (scores 0–2) and high invasion (3–4), a decision taken before performing survival analyses. We assessed the inter-reliability of the invasion score binary classification between two scorers using the kappa statistic, which showed 0.7 concordance for the C cohort (weighted kappa = 0.7, 95% CI = 0.64–0.77).

The amount of collagen at the IF of the tumour and in tumour-adjacent skin was scored from H&E slides (C cohort) according to abundance of distinctly formed collagen bundles. Two-independent pathologists examined collagen on H&E routine-stained sections of normal skin surrounding the melanomas and the collagen adjacent/enveloping the IF of the tumour in the dermis at 100–200× magnification. The following scoring system was used: collagen absent or low (1): when fully formed collagen bundles were rare, and the visible collagen was distributed in haphazard smaller fragments or unidentifiable in an amorphous deposit of elastotic material. Low collagen (2): when well-defined, undulating fibres of normal dermal length collagen, are scarce, and a pattern of elastotic (fragmented or aggregate) material predominates. Medium collagen (3): well-defined, undulating and organised fibres coexist with aggregate elastotic material. High collagen (4): well-defined fibres in organised disposition predominating, with minimal or absent elastotic material interspersed between the tight bundles (Supplementary Fig. 4m). We assessed the inter-reliability of the collagen score classification between two scorers using the kappa statistic, which showed 0.78 concordance for the C cohort (weighted kappa = 0.78, 95% CI = 0.7–0.81). A subset of samples in cohort A (n = 16), for which tissue was available, were stained with trichrome of Masson to quantify collagen. We were unable to score the B cohort IF due to Covid-19 international restrictions.

**Statistical analysis**. Data collection was performed with Microsoft Excel 2010 (64-bit, Microsoft). For in vitro studies, statistical analysis was performed in GraphPad Prism (version 7.01, GraphPad Software, Inc.). For comparisons between two groups, Mann–Whitney tests were used and for comparisons between multiple groups, Kruskal–Wallis with Dunn's multiple comparison tests. A $p$ value < 0.05 was considered significant, after correcting for multiple testing where necessary. For human studies, statistical analysis was performed in R (version 3.5.1, RStudio v1.2.5001, RStudio Inc.). Association between categorical data was performed with Fisher exact tests. Survival analysis was performed using survival (version 3.1–12) and survminer (version 0.4.6) packages. For all clinical cohorts, MSS, overall survival (OS) and PFS were calculated from time of diagnosis. Univariate grouped survival analysis performed with Kaplan–Meier and log-rank tests, and multi-variate analyses with Cox regression models, with evaluation of the proportional hazard assumption. Gene expression ($\log_2(x+1)$ normalised RSEM) and clinical

data from the TCGA SKCM and PANCAN data sets was accessed from the UCSC Xena data portal (https://xenabrowser.net/datapages/). Samples were grouped into COL1A1 high or low based on the expression relative to the median expression of all samples. The MAF score for each sample was determined by calculating the geometric mean of all genes in a published MAF signature[37]. High and low MAF samples were classified based on the median signature score.

**Reporting summary**. Further information on research design is available in the Nature Research Reporting Summary linked to this article.

## Data availability
The RNA-seq data used in this study has been previously published and available from ENA project PRJEB13731. The remaining data are available within the article, Supplementary Information or available from the authors upon reasonable request. Source data are provided with this paper.

## Code availability
No custom codes or algorithms were used in the analysis of this study, and all analyses packages and software are cited in the manuscript. R codes used are available from the corresponding author upon request.

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

## Acknowledgements

A.V. is a Wellcome Beit Fellow and personally funded by a Wellcome Trust Intermediate Fellowship (110078/Z/15/Z), and has work funded by Cancer Research UK (A27412) and Leo Pharma Foundation. E.N. is funded by Fondo de Investigación en Salud (FIS) PI15/01860, Instituto Carlos III, Spain. C.G.-M. funded by the French Dermatology Society, Collège des Enseignants en Dermatologie de France (CEDEF) and UNICANCER France. S.J.F. acknowledges support from the European Commission (FP7-PEOPLE-2013-IEF—6270270) and the Royal College of Surgeons in Ireland StAR programme. P.M. is funded by the Manchester Cancer Research Centre, supported by the CRUK Manchester Institute. We acknowledge the generous contribution of Christie UK Biobank, APHM Biobank (France) and IVO Biobank (Spain). Bioresources were provided by the Biological Resources Centre of the Assistance Publique Hôpitaux de Marseille, (CRB APHM, certified NF S96-900 and ISO 9001 v2015), from the CRB-TBM component (BB-0033-00097), from the Biobank of the Instituto Valenciano de Oncología, Valencia, Spain, and from the Christie Biobank. We thank Prof. Tim Somervaille and Prof. Iain Hagan, of Cancer Research UK, for critical review of the manuscript. We thank Cancer Research UK Manchester Institute Facilities, in particular histology (Garry Ashton and Caron Behan), imaging (Kang Zeng and Steve Bagley), sequencing (Dr. Wolfgang Breitwieser) and research integrity (Dr. Andrew Porter). We thank the University of Manchester BioAFM Lab Core Facility (Dr. Nigel Hodson).

## Author contributions

Conceptualisation: A.V.; writing of the original manuscript and generation of figures: T.B. and A.V.; collection and analysis of data: T.B., A.V., E.N., C.G.-M., L.M., V.T., A.P., A.M., E.K., S.Z., S.G, C.H.E. and S.J.F.; tools and methods: T.B., A.V., E.N., A.P., A.M., E.K., S.Z., S.G., S.J.F., J.K., P.M., K.R. and S.C. Funding acquisition, project administration, supervision: A.V. Resources: J.K., E.N., S.J.F. and S.Z. Software: T.B., E.K., S.Z. and S.J.F.

## Competing interests

The authors declare no competing interests.
