## [Peer Review File · Nature Communications]

REVIEWER COMMENTS

Reviewer #1 (Remarks to the Author):

In this study the authors show that UV damage in aged skin can actually inhibit metastatic progression of melanoma cells, contrary to studies of non-UV damaged aged skin that actually show the increased invasion of melanoma cells with age. Overall the manuscript is interesting, with some elegant experiments, but several details are omitted, that should be included in the body of the text.

1) The introduction should be expanded to encompass more details. First, it would be useful to explain that melanoma outcome in the elderly follows a bell curve, where prognosis is poorer for patients aged 65-80 or so, then starts to improve again for very elderly patients. The Shenoy lab has shown that migration of melanoma in dense or soft matrices similarly follows a bell curve (see Azhamadzeh et al, PNAS), almost identical to the panels shown in Figure 2D, where maximal invasion occurs at ~1.5mg/mL of collagen. Work from the Weaver lab should also be discussed in the introduction.

2) Patient ages are included in the extended data, but should be more specifically alluded to in the text, and with each experiment they are being used. It is unclear whether the fibroblasts used in different assays are foreskin fibroblasts which have been UV irradiated, or fibroblasts from young, non-sun exposed patient skin that have been irradiated, or whether the patient fibroblasts are used "as is" with the sun-exposed coming from older, sun-exposed patients.

3) For the skin reconstructs, especially in figure 3, it would be good to have zoomed out fields of view so the overall invasion along the reconstruct can be evaluated. 2-photon microscopy of the collagen in the reconstructs should also be performed in this figure to determine if there are fundamental differences in the collagen structure with MMP1.

4) The inhibition experiments require a little more rigor. Batimastat is a broad spectrum MMP inhibitor, inhibiting MMPs 1,2, 9 and 7 in roughly the same IC50 range, and only 1 shRNA was used (it appears) against MMP1. Further to determine if the effect is dependent solely on MMP1, control experiments with MMP2 or 9 shRNAs would be a nice addition.

5) More ECM specific studies in the absence and presence of MMP1 would also lend more support to this paper. Quantitative 2-photon microscopy of collagen in the skin reconstructs in Figure 3g for example, and trichrome studies (such as those in Extended Data 4) would support these experiments. Assessment of other ECM proteins, eg elastin and fibronectin would also be important to validate that techniques being used are indeed producing elastosis and other similar UV-related phenotypes.

Reviewer #2 (Remarks to the Author):

In the manuscript "Ultraviolet light-induced collagen degradation inhibits melanoma invasion" the authors investigate the effects of ultraviolet radiation on dermal fibroblast remodeling of the extracellular matrix in cutaneous melanoma. Specifically, they find a decrease in collagen in areas of sun damaged skin as compared to sun protected skin, which is also associated with changes in gene expression of fibroblasts in the dermis, including down-regulation of collagen genes and upregulation of matrix metalloproteinases (MMPs). Furthermore, the effects of sun exposure on MMP expression, specifically upregulation of MMP-1, can be recreated in vitro by exposing fibroblasts to ultraviolet radiation (UVR). The authors also find that collagen concentration can regulate melanoma cell invasion in an in vitro spheroid model, specifically that very low or very high concentrations of collagen reduce invasion in multiple melanoma cell lines. The effect of MMP degradation on collagen concentration and reduced melanoma invasion is characterized through a comprehensive set of in vitro experiments, through enzymatic collagen degradation, by UVR exposed fibroblasts, or by reducing MMP activity with a pharmacological inhibitor or through shRNA knockdown of MMP-1 in fibroblasts. Primary cutaneous melanoma samples were examined for collagen and single cell invasion, finding less collagen was associated with fewer invading cells and improved melanoma specific survival. Finally, the authors expanded their analysis beyond melanoma, and found expression of COL1A1 in a number of solid tumors was associated with a greater risk of death and shorter progression free survival. This is an interesting and important

work that offers some explanation for the contradictory effects of UVR on patient survival and contributes to the literature that physical aspects of the tumor microenvironment are key contributors to tumor invasion. This work also identifies COL1A1 as a potentially powerful biomarker of poor patient outcomes in a wide array of solid tumors. The work is well described and supported by the data presented, methods described, and statistical analysis using both in vitro mechanistic experiments and primary human samples. Several minor comments may improve the manuscript:

1. References on previous use of AFM imaging to analyze characteristics of the fibroblast produced matrices should be provided in text and/or methods. Standard characterization of these matrices often involves immunostaining of fibronectin or collagen, and then computational analysis (i.e. CT-FIRE) which would provide measurements of fiber length and organization referred to in the manuscript. "... atomic force microscopy (AFM) topographic imaging revealed UV-HFF fibroblast-generated ECM presented more fragmented, sparser and disorganized matrix fibrils than UVR-naïve, HFF fibroblasts (Fig. 1c, d)." It's not very clear that the AFM images presented support claims of fragmented or aligned/disorganized fibers - the wording should be toned down or the matrices analyzed in another manner.

2. The method and quantification for the collagen degradation assay using DQ collagen is unclear. Are the authors using a decrease or increase in fluorescent signal to indicate degradation? DQ is dye quenched collagen, highly labeled collagen in which the FITC self quenches and upon degradation the quenching is relieved and the fluorescent signal increases (as described by the manufacturer). Why was the area of signal used as opposed to intensity? It is confusing that the figure is labeled with FITC collagen as opposed to DQ collagen. Representative images that were used for the different scoring of low, medium, and high in the supplemental would be helpful. It is also possible changes in fluorescence could be more quantitatively measured using a plate reader.

Reviewer #3 (Remarks to the Author):

The main goal of the studies described here is to examine whether "UVR modifies dermal fibroblast function". The results confirmed some published information, such as reduction in collagen in UVR-damaged dermis and the presence of high levels of somatic mutations in fibroblasts adjacent to the tumor of sun-damaged dermis. The novel data, in a way paradoxical (as expressed also by the authors) are that "UVR damage to the dermis destroys collagen, limiting invasion and improving outcome". They showed that the invasion of melanoma cells is optimal at specific collagen concentration (1.5 mg/ml), and that degraded collagen limit melanoma invasion, i.e., less collagen less invasion and more collagen more invasion. They expanded the studies and showed that COL1A1 expression is a "biomarker for primary pan-cancer survival, in which young and aged patients with multiple cancers expressing high levels of COL1A1 are at greater risk of death and have shorter progression free survival".

The results are based on elaborate experiments that support the conclusions.

The manuscript is written in somewhat repetitious and confusing manner. It is sometime not clear if the authors talk about their own results or published information, that turned out to appear as a reference, especially in the abstract.

Examples:

"The expression of collagen-cleaving matrix metalloprotein-1 (MMP1) by UVR-damaged fibroblasts was persistently upregulated to reduce local levels of collagen 1 (COL1A1)¹⁰."

"We characterized gene expression in fibroblasts obtained from matched UVR-damaged and UVR-protected dermis from healthy donors¹⁸."

**REVIEWER COMMENTS –
Ultraviolet light-induced collagen degradation inhibits melanoma invasion**

Reviewer #1 (Remarks to the Author):

In this study the authors show that UV damage in aged skin can actually inhibit metastatic progression of melanoma cells, contrary to studies of non-UV damaged aged skin that actually show the increased invasion of melanoma cells with age. Overall the manuscript is interesting, with some elegant experiments, but several details are omitted, that should be included in the body of the text.

1) The introduction should be expanded to encompass more details. First, it would be useful to explain that melanoma outcome in the elderly follows a bell curve, where prognosis is poorer for patients aged 65-80 or so, then starts to improve again for very elderly patients. The Shenoy lab has shown that migration of melanoma in dense or soft matrices similarly follows a bell curve (see Azhamadzeh et al, PNAS), almost identical to the panels shown in Figure 2D, where maximal invasion occurs at ~1.5mg/mL of collagen. Work from the Weaver lab should also be discussed in the introduction.

Response 1

Thank you for the overall assessment of our work.

We agree there are two categories of aged patients and now include this in our introduction: the aged and the superaged, who have unique disease incidence and progression, with decreasing rate of disease and death in the “superaged”. This is important because it highlights the link between UV damage and melanoma is not linear.

We also correct the previous omission of seminal Weaver and Shenoy work, which have now enhanced our introduction. We add to the previous description of the link between collagen and melanoma/cancer cell behaviour, discussing the effect of collagen likely extends beyond a pure scaffold function and is multifaceted.

Page 2, Lines 10-21:

“UVR damage accumulates with increasing decades of life, and aged patients have worse melanoma survival^{17,18}. Therefore, it is possible that chronic UVR damage may lead to shorter melanoma specific survival. However, in common with some non-hormonal cancers, the incidence and mortality of melanoma sharply rise after age 60, and then significantly decrease after age 85^{19,20}, suggesting the relationship between cumulative UVR exposure, cutaneous damage, age and melanoma death is not linear.

Previous studies have shown collagen quantity in the extracellular matrix modifies melanoma cell behaviour²¹. Surprisingly, both increased²² and decreased²³ deposition of collagen have been linked to malignant behaviour, suggesting the effect of collagen on cancer behaviour extends beyond protein level and scaffold function. In this study we explore how collagen levels in the dermis, which vary according to sun damage and age, affect melanoma survival.”

References

17. National Cancer Institute. The Surveillance, Epidemiology, and End Results (SEER) Program. Available at: www.seer.cancer.gov. Accessed June 2019.
18. Cancer Research UK. <https://www.cancerresearchuk.org/health-professional/cancer-statistics/statistics-by-cancer-type/melanoma-skin-cancer/mortality#heading-One>
19. Hashim, D. *et al.* Cancer mortality in the oldest old: a global overview. *Aging (Albany, NY)*. **12**, 16744–16758 (2020).
20. Stanta, G., Campagner, L., Cavallieri, F. & Giarelli, L. Cancer of the oldest old. What we have learned from autopsy studies. *Clin. Geriatr. Med.* **13**, 55–68 (1997).
21. Ahmadzadeh, H. *et al.* Modeling the two-way feedback between contractility and matrix realignment reveals a nonlinear mode of cancer cell invasion. *Proc. Natl. Acad. Sci. U. S. A.* **114**, E1617–E1626 (2017).
22. Levental, K. R. *et al.* Matrix crosslinking forces tumor progression by enhancing integrin signaling. *Cell* **139**, 891–906 (2009).
23. Arnold, S. A. *et al.* Lack of host SPARC enhances vascular function and tumor spread in an orthotopic murine model of pancreatic carcinoma. *Dis. Model. Mech.* **3**, 57–72 (2010).

2) Patient ages are included in the extended data, but should be more specifically alluded to in the text, and with each experiment they are being used. It is unclear whether the fibroblasts used in different assays are foreskin fibroblasts which have been UV irradiated, or fibroblasts from young, non-sun exposed patient skin that have been irradiated, or whether the patient fibroblasts are used “as is” with the sun-exposed coming from older, sun-exposed patients.

Response 2

Thank you for pointing this out. We have now clarified each model origin and ages throughout the manuscript. We now specify the origin of the cell used for each panel and age in the text, figures, figure legends and methods. We also specify whether patient-derived fibroblasts received additional UV. All the edits are highlighted across the new manuscript.

3) For the skin reconstructs, especially in figure 3, it would be good to have zoomed out fields of view so the overall invasion along the reconstruct can be evaluated. 2-photon microscopy of the collagen in the reconstructs should also be performed in this figure to determine if there are fundamental differences in the collagen structure with MMP1.

Response 3

We agree this improves the quality of the data and have addressed your points. We now provide lower magnification fields that better depict the phenomenon we are trying to quantify, as well as multiple new images from our models in the extended data which add visual scope to our models.

For the photomicrographs we provide higher scope magnification in Extended Data Figure 2, and we have particularly, as requested, provided expanded views for Extended Data Figure 3. These are referenced in the main text and described in new legends.

Extended Data Figure 2C

Legend. (c) Representative H&E photomicrographs of UV and UV-HFF constructs with melanoma cells (scale bar 150 μ m)

Extended Data Figure 3M

Legend. Representatives H&E photomicrographs of shCtrl-HFF (four top left images), shCtrl-UV-HFF (four top right images), shMMP1-HFF (four bottom left images) and shMMP1-UV-HFF (four bottom right images) derived ECM constructs with invading melanoma cells, scale bars 100 μ m.

Additionally, we provide second harmonic generation imaging of collagen constructs that reveal differences in the UV/non-UV matrices. Specifically, we show matrices generated by UV-damaged fibroblasts contain significantly fewer collagen fibres, and reassuringly, knocking out MMP1 with shRNA reverses the phenotype. These data have been added in Extended Data Fig. 2d, discussed in the main text and methods. (We add further additional work of our constructs to Response 5).

Extended Data Figure 2D, E

Legend. (d) Second harmonic generation (SHG) imaging of collagen fibres in organotypic dermal collagen HFF and UV-HFF constructs, scale bar = 50 μm . **(e)** Quantification of collagen from SHG images in HFF and UV-HFF, (Mann Whitney U **** $p < 0.0001$).

Page 5, Lines 7-11:

“UV-HFF constructs replicated the cardinal features of UVR damage³³⁻³⁵, with significantly reduced collagen levels compared to HFF constructs ($p < 0.0001$, Extended Data Fig. 2d, e). Additionally, UV-HFF constructs presented reduced fibronectin (Extended Data Fig. 2f), and no difference in elastin expression compared to HFF constructs (Extended Data Fig. 2e).”

4) The inhibition experiments require a little more rigor. Batimastat is a broad spectrum MMP inhibitor, inhibiting MMPs 1,2, 9 and 7 in roughly the same IC50 range, and only 1 shRNA was used (it appears) against MMP1. Further to determine if the effect is dependent solely on MMP1, control experiments with MMP2 or 9 shRNAs would be a nice addition.

Response 4

We agree inhibitory experiments are required to improve inhibition experiments. The initial lentiviral transfection of shMMP1 contained 3-5 constructs targeting *MMP1*. We have now performed validation experiments with an additional siRNA targeting MMP1, replicating the initial findings. Additionally, we have performed as suggested the experiment with an shRNA lentivirus targeting MMP2. Our new results show that knockdown of MMP2 does not affect collagen degradation in UV-fibroblasts, lending further support to the key role in collagen degradation of MMP1. Because our HFF cell line model does not express MMP9 in detectable levels, we did not pursue knockdown experiments with MMP9. These expanded results are now added to the manuscript in Extended Data Fig 3, discussed in the main text and methods.

Extended Data Fig 3B, D-I

Legend. (b) Western blots validating knockdown of MMP1 and MMP2 in shRNA cell lines. (d) Representative images of collagen degradation in siCtrl-HFF, siCtrl-UV-HFF, siMMP1-HFF and siMMP1-UV-HFF fibroblasts. Green: intact DQ collagen; red: phalloidin; blue: Hoechst. Size bars: 20 μ m. (e) Validation of siRNA effect on *MMP1* relative expression (RE) by qPCR in collagen degradation assay. (f) Quantification of collagen degradation of siCtrl-HFF and siMMP1-HFF (pink) and their isogenic chronic UVR cell lines siCtrl-UV-HFF and siMMP1-UV-HFF (blue), (Mann Whitney U **** p <0.0001). (g) Representative images of collagen degradation in shMMP2-HFF, and shMMP2-UV-HFF fibroblasts. Green: intact DQ collagen; red: phalloidin; blue: Hoechst. Size bars: 20 μ m. (h) Quantification of collagen degradation of shMMP2-HFF and their isogenic chronic UVR cell line shMMP2-UV-HFF (blue), (Mann Whitney U **** p <0.0001). (i) Quantification of fibres within 10° of mode orientation in shCtrl-HFF, shCtrl-UV-HFF, shMMP1-HFF and shMMP1-UV-HFF derived ECM by fibronectin immunofluorescence (shCtrl HFF vs shCtrl UV-HFF p =0.1 (n=3), ns: not significant).

Page 7, Line 14-19

“The specific role of MMP1 was validated with an additional knockdown with an siRNA targeting MMP1 (Extended Data Fig. 3d, e, f). Additionally, we found shRNA targeting shMMP2 did not modify collagen degradation (Extended Data Fig. 3g, h); and knockdown of MMP1 restored the alignment of fibres in UV-HFF matrices. (Fig. 3g, h, Extended Data Fig. 3i).”

5) More ECM specific studies in the absence and presence of MMP1 would also lend more support to this paper. Quantitative 2-photon microscopy of collagen in the skin reconstructs in Figure 3g for example, and trichrome studies (such as those in Extended Data 4) would support these experiments. Assessment of other ECM proteins, eg elastin and fibronectin would also be important to validate that techniques being used are indeed producing elastosis and other similar UV-related phenotypes.

Response 5

We now provide improved images for all constructs including the MMP1 inhibitor experiment (above, response 3). Our constructs did not withstand trichrome staining, so we performed second harmonic generation imaging of the collagen that reveal differences in the UV/non-UV matrices. Specifically, we show constructs with UV-

HFF contain significantly fewer collagen fibres, with the knockout of MMP1 with shRNA reversing this phenotype. These data are now added in Extended Data Fig. 2, 3; as well as discussed in the main text.

Additional data has been added as well, showing elastin expression between HFF and UV-HFF and fibronectin levels in collagen constructs for both HFF/UV-HFF and shMMP1-HFF/shMMP-UV-HFF. Our model mimics human photodamage, as it shows UV driven reduction in fibronectin, which is then restored upon MMP1 knockdown. Our results also concur with other studies that show elastin quantity is not diminished following UV³⁸. We hope this improves the quality of results by providing additional features of the UV phenotype. We have added the new data to the figures, main text, figure legends and methods.

Reference 38:

Schwartz, E., Feinberg, E., Lebowhl, M., Mariani, T. J. & Boyd, C. D. Ultraviolet radiation increases tropoelastin accumulation by a post-transcriptional mechanism in dermal fibroblasts. *J. Invest. Dermatol.* **105**, 65–69 (1995).

Figure 1E-G

Legend. (e) Quantification of fibre alignment distribution in human foreskin fibroblasts HFF and UV-HFF derived ECM by fibronectin immunofluorescence (f) Fraction of fibres within 10° of mode orientation, Mann-Whitney U **p<0.01 (g) Immunofluorescence of fibronectin fibres in decellularised HFF and UV-HFF derived ECM, colour coded for orientation of fibre, cyan represents mode, scale bar: 25 µm.

Page 4, Line 4-7:

“Furthermore, immunofluorescent staining of fibronectin fibres in HFF and UV-HFF derived ECM, confirmed that UV-HFF matrices were significantly more disorganised with fewer aligned fibres than matrices generated by HFF fibroblasts (Fig. 1e-g).”

Extended Data Figure 2D-G

(d) Second harmonic generation (SHG) imaging of collagen fibres in organotypic dermal collagen HFF and UV-HFF constructs, scale bar = 50 μ m. **(e)** Quantification of collagen from SHG images in HFF and UV-HFF, (Mann Whitney U **** $p < 0.0001$). **(f)** Fibronectin IHC staining in organotypic dermal collagen HFF and UV-HFF constructs. **(g)** Label free quantification (LFQ) of elastin (ELN) in the HFF and UV-HFF matrix by mass spectrometry (ns: not significant).

Page 5, Line 7-11:

“UV-HFF constructs replicated the cardinal features of UVR damage^{33–35}, with significantly reduced collagen levels compared to HFF constructs ($p < 0.0001$, Extended Data Fig. 2d, e). Additionally, UV-HFF constructs presented reduced fibronectin (Extended Data Fig. 2f), and no difference in elastin expression compared to HFF constructs (Extended Data Fig. 2g).”

Extended Data Figure 3J-L

Legend. **(j)** Representative H&E images of shMMP1-HFF and shMMP1-UV-HFF fibroblasts, scale bars 75 μ m (left) and second harmonic generation (SHG) imaging of collagen fibres in organotypic dermal collagen shMMP1-HFF and shMMP1-UV-HFF constructs (right), scale bars 50 μ m. **(k)** Fibronectin IHC staining in organotypic dermal collagen shMMP1-HFF and shMMP1-UV-HFF constructs, scale bars 50 μ m. **(l)** Quantification of collagen from SHG images in shMMP1-HFF and shMMP1-UV-HFF (Mann Whitney U, ** $p < 0.01$).

Page 7, Line 22-24:

“Knockout of MMP1 restored collagen and fibronectin levels in UV-HFF constructs to similar levels as HFF (Extended Data Fig. 3j, k, l).”

Reviewer #2 (Remarks to the Author):

In the manuscript “Ultraviolet light-induced collagen degradation inhibits melanoma invasion” the authors investigate the effects of ultraviolet radiation on dermal fibroblast remodeling of the extracellular matrix in cutaneous melanoma. Specifically, they find a decrease in collagen in areas of sun damaged skin as compared to sun protected skin, which is also associated with changes in gene expression of fibroblasts in the dermis, including down-regulation of collagen genes and upregulation of matrix metalloproteinases (MMPs). Furthermore, the effects of sun exposure on MMP expression, specifically upregulation of MMP-1, can be recreated in vitro by exposing fibroblasts to ultraviolet radiation (UVR). The authors also find that collagen concentration can regulate melanoma cell invasion in an in vitro spheroid model, specifically that very low or very high concentrations of collagen reduce invasion in multiple melanoma cell lines. The effect of MMP degradation on collagen concentration and reduced melanoma invasion is characterized through a comprehensive set of in vitro experiments, through enzymatic collagen degradation, by UVR exposed fibroblasts, or by reducing MMP activity with a pharmacological inhibitor or through shRNA knockdown of MMP-1 in fibroblasts. Primary cutaneous melanoma samples were examined for collagen and single cell invasion, finding less collagen was associated with fewer invading cells and improved melanoma specific survival. Finally, the authors expanded their analysis beyond melanoma, and found expression of COL1A1 in a number of solid tumors was associated with a greater risk of death and shorter progression free survival. This is an interesting and important work that offers some explanation for the contradictory effects of UVR on patient survival and contributes to the literature that physical aspects of the tumor microenvironment are key contributors to tumor invasion. This work also identifies COL1A1 as a potentially powerful biomarker of poor patient outcomes in a wide array of solid tumors. The work is well described and supported by the data presented, methods described, and statistical analysis using both in vitro mechanistic experiments and primary human samples. Several minor comments may improve the manuscript:

1. References on previous use of AFM imaging to analyze characteristics of the fibroblast produced matrices should be provided in text and/or methods. Standard characterization of these matrices often involves immunostaining of fibronectin or collagen, and then computational analysis (i.e. CT-FIRE) which would provide measurements of fiber length and organization referred to in the manuscript. “... atomic force microscopy (AFM) topographic imaging revealed UV-HFF fibroblast-generated ECM presented more fragmented, sparser and disorganised matrix fibrils than UVR-naïve, HFF fibroblasts (Fig. 1c, d).” It’s not very clear that the AFM images presented support claims of fragmented or aligned/disorganized fibers - the wording should be toned down or the matrices analyzed in another manner.

Response 1

Thank you for your assessment of our work.

We now have appropriate references for AFM imaging analysis in the methods. We thank the reviewer for suggesting computational improvement and rewording, which we have accomplished by adding further details on the analysis of the AFM roughness value and how it relates to the structure of the ECM in the hope of clarifying this. This has been added to both the results section of the manuscript as well as the methods with appropriate references.

In the Methods, Page 5, Lines 1-5:

“Roughness is the root mean square average of the image and is calculated based on the height difference per pixel along the sample length. Rq is used to study the surface topography of various nanostructures^{21,22}. Rq provides a quantitative measure of fibril organisation in dermis and could possibly suggest the integrity of matrix²³.”

References:

- 21 Webb, H. K. *et al.* Roughness parameters for standard description of surface nanoarchitecture. *Scanning* **34**, 257-263, doi:10.1002/sca.21002 (2012).
- 22 Girasole, M. *et al.* Roughness of the plasma membrane as an independent morphological parameter to study RBCs: a quantitative atomic force microscopy investigation. *Biochimica et biophysica acta* **1768**, 1268-1276, doi:10.1016/j.bbamem.2007.01.014 (2007).
- 23 Argyropoulos, A. J. *et al.* Alterations of Dermal Connective Tissue Collagen in Diabetes: Molecular Basis of Aged-Appearing Skin. *PloS one* **11**, e0153806, doi:10.1371/journal.pone.0153806 (2016).

In addition, we have now performed immunofluorescent staining and microscopy of fibronectin on the fibroblast derived ECM and performed computational analysis of fibre orientation as a second method to validate that the UV-HFF fibroblasts produce a more disorganised matrix structure. Because of the key role of fibronectin in cutaneous homeostasis we have included this computation to the manuscript (see response to Reviewer 1, Figure 1E-G).

In the Main text, we have reworded the initial description and added the computation, Page ,3 Lines 24-26) and Page 4, Lines 1-5:

“Additionally, atomic force microscopy (AFM) topographic imaging suggested UV-HFF fibroblast-generated ECM presented more fragmented, sparser and disorganised matrix fibrils than UVR-naïve, HFF fibroblasts. The higher roughness (Rq) value indicates less symmetry across the ECM surface plane, in keeping with degradation of UV-HFF fibroblast-generated ECM³⁰⁻³² (Fig. 1c, d). Furthermore, immunofluorescent staining of fibronectin fibres in HFF and UV-HFF derived ECM, confirmed that UV-HFF matrices were significantly more disorganised with fewer aligned fibres than matrices generated by HFF fibroblasts (Fig. 1e-g).”

2.The method and quantification for the collagen degradation assay using DQ collagen is unclear. Are the authors using a decrease or increase in fluorescent signal to indicate degradation? DQ is dye quenched collagen, highly labeled collagen in which the FITC self quenches and upon degradation the quenching is relieved and the fluorescent signal increases (as described by the manufacturer). Why was the

area of signal used as opposed to intensity? It is confusing that the figure is labeled with FITC collagen as opposed to DQ collagen. Representative images that were used for the different scoring of low, medium, and high in the supplemental would be helpful. It is also possible changes in fluorescence could be more quantitatively measured using a plate reader.

Response 2

Thank you for highlighting this unclear method description. We have now corrected our explanations and relabelled to avoid confusion. The reviewer is correct, we use loss of collagen signal to quantify in the perimeter of the fibroblasts how much has been degraded, and this is hopefully now made clear in the updated manuscript and labels. We used complete loss- i.e no signal, precisely due to the limitations of the opposite quantification, where fluorescent signalling increases upon degradation. To better explain this, we have extended the method and provide images of the different fibroblast scenarios we scored. Two blinded scorers assessed images from 3 biological replicates of UV treatments and isogenic lines throughout the experiments.

Extended Data Figure 3A

Legend: (a) Representative images of collagen degradation scores as outlined in methods.

Reviewer #3 (Remarks to the Author):

The main goal of the studies described here is to examine whether “UVR modifies dermal fibroblast function”. The results confirmed some published information, such as reduction in collagen in UVR-damaged dermis and the presence of high levels of somatic mutations in fibroblasts adjacent to the tumor of sun-damaged dermis. The novel data, in a way paradoxical (as expressed also by the authors) are that “UVR damage to the dermis destroys collagen, limiting invasion and improving outcome”. They showed that the invasion of melanoma cells is optimal at specific collagen concentration (1.5 mg/ml), and that degraded collagen limit melanoma invasion, i.e., less collagen less invasion and more collagen more invasion. They expanded the studies and showed that COL1A1 expression is a “biomarker for primary pan-cancer survival, in which young and aged patients with multiple cancers expressing high levels of COL1A1 are at greater risk of death and have shorter progression free survival”.

The results are based on elaborate experiments that support the conclusions.

The manuscript is written in somewhat repetitious and confusing manner. It is sometime not clear if the authors talk about their own results or published information, that turned out to appear as a reference, especially in the abstract.

Examples:

“The expression of collagen-cleaving matrix metalloprotein-1 (MMP1) by UVR-damaged fibroblasts was persistently upregulated to reduce local levels of collagen 1 (COL1A1)¹⁰.”

“We characterized gene expression in fibroblasts obtained from matched UVR-damaged and UVR-protected dermis from healthy donors¹⁸.”

Response 1

Thank you for your review of our data and for highlighting the key findings are supported by the experiments.

We welcome the opportunity to more clearly separate previous publications from our own data, and have attempted to clarify this throughout, with particular care in the abstract.

Our new abstract specifies our findings underlined:

“Ultraviolet radiation (UVR) increases the incidence of cutaneous melanoma¹⁻⁴. The ageing, sun-exposed dermis accumulates UVR damage⁵, and older patients develop more melanomas at UVR-exposed sites^{4,6,7}. As fibroblasts are functionally heterogeneous and play key roles in the stromal contribution to cancer^{8,9}, we asked whether UVR modifies dermal fibroblast function. Here we confirmed the expression of collagen-cleaving matrix metalloprotein-1 (MMP1) by UVR-damaged fibroblasts was persistently upregulated to reduce local levels of collagen 1 (COL1A1), and found dermal COL1A1 degradation by MMP1 decreased melanoma invasion. Conversely, we show inhibiting extracellular matrix degradation and MMP1 expression restored melanoma invasion to UVR damaged dermis. We confirmed *in vitro* findings in a cohort of primary cutaneous melanomas of aged humans, showing more cancer cells invade as single cells at the invasive front of melanomas expressing and depositing more collagen. We found collagen and single melanoma cell invasion are robust predictors of poor melanoma-specific survival. These data indicate melanomas arising over UVR-damaged, collagen-poor skin of the elderly are less invasive, and this reduced invasion improves survival. Consequently, although UVR increases tumour incidence, it delays primary melanoma invasion by degrading collagen. However, we show melanoma-associated fibroblasts can restore invasion in low-collagen primary tumours by increasing collagen synthesis. Finally, we demonstrate high COL1A1 gene expression is a biomarker of poor outcome across a broad range of primary cancers.”

In page 3, lines 6-9:

“We analysed gene expression in human adult fibroblasts to compare matched UVR-damaged and UVR-protected dermis from healthy donors (median age 42, range 19-66²⁵) “.

An additional place where we have clarified this is in the Discussion, Page 10, line 3:

“We confirmed our *in vitro* results, showing that in aged primary cutaneous melanomas, single tumour cells invading the dermis and collagen at the IF robustly predict poor survival.”

Reviewer#4 (Remarks to the Author):

1. For the figure 4 survival analysis, please provide clear description about the

definition of the starting time point for calculating the length of PFS, MSS or OS for each clinical cohort and TCGA cohort.

Response 1

We have made this clarification to improve the results. We have specified the starting points in the methods and in the key figure legends.

In the Methods, Page 9, Lines 7-9:

“For all clinical cohorts, melanoma specific survival (MSS), overall survival (OS) and progression free (PFS) were calculated from time of diagnosis.”

2. Please provide the number of patients at risk at the beginning of follow-up in each level of compared categorical group. Such information will reflect the sample size of each subgroup, which may be used by readers to evaluate the reliability of findings.

Response 2

We now provide risk tables in the new Extended Data Table 6. These data accurately outline the number and percent of patients at risk for all Kaplan Meier analyses performed in Figure 4 and Extended Data Figure 4.

3. In addition, in the section of Histological and clinical sample analysis, it will be informative if authors can describe in detail about what the cut-points are based on the scores from H&E slides, and how were they determined for the studied biomarkers, e.g. how to separate the four level of collagen quantity (fig.4f), as well as high vs. low of COL1A1 or MAF (fig. 4g, j).

Response 3

We agree improving the level of information on quantification will facilitate validation and support development of robust biomarkers of outcome, that can be tested in other cancers. We include now a more finely tuned description, kappa interobserver agreements and representative clinical images within the methods to address this in depth.

For solar elastosis, in the Methods, Page 9, Line 20-27:

“Landi et al established a scoring system for the degree of solar elastosis from absent to severe using an 11-point score, from 0 to 3+. To generate binary categories, cases are classified as bearing chronic sun damage (CSD), for scores between 0 to 2-, or non-CSD for scores 2 to 3+. Cutoffs for absent (range 0, 0+), low (range 1- to 1+), moderate (range 2- to 2+) and high (3- to 3+) were established from the same range, as previously used²⁶. We assessed the inter- reliability of the binary CSD classification between 2 scorers using the kappa statistic, which showed 0.75 concordance for the B cohort (weighted kappa= 0.75, 95% CI= 0.69-0.79).“

Reference:

²⁶ Moreno, A. *et al.* Histologic Features Associated With an Invasive Component in Lentigo Maligna Lesions. *JAMA dermatology* **155**, 782-788, doi:10.1001/jamadermatol.2019.0467 (2019).

For the collagen quantification, in Methods, Page 10, Line 7-21:

“The amount of collagen at the IF of the tumour and in tumour-adjacent skin was scored from H&E slides (C cohort) according to abundance of distinctly formed collagen bundles. Two independent pathologists examined collagen on H&E routine stained sections of normal skin surrounding the melanomas and the collagen adjacent/enveloping the invasive front of the tumour in the dermis at 100-200x magnification. The following scoring system was used: collagen absent or low (1): when fully formed collagen bundles were rare, and the visible collagen was distributed in haphazard smaller fragments or unidentifiable in an amorphous deposit of elastotic material. Low collagen (2): when well-defined, undulating fibres of normal dermal length collagen are scarce, and a pattern of elastotic (fragmented or aggregate) material predominates. Medium collagen (3): well-defined, undulating and organised fibres coexist with aggregate elastotic material. High collagen (4): well-defined fibres in organised disposition predominating, with minimal or absent elastotic material interspersed between the tight bundles. We provide representative images below. We assessed the inter- reliability of the collagen score classification between 2 scorers using the kappa statistic, which showed 0.78 concordance for the C cohort (weighted kappa= 0.78, 95% CI= 0.7-0.81).”

Collagen 4

Collagen 3

Collagen 2

Collagen 1

Methods Figure 1. Representative photomicrographs of collagen scoring. 4: preservation of collagen fibres in the dermis. 3. Combination of preserved, normal collagen bundles (pink) intermixed with elastotic fibres of non-collagenous material (purple). 2. Combination of some collagen (pink) with purple elastotic fibres. Multiple fragmentation of collagen. 1. Scarce or absent collagen, substitution of the matrix by elastotic heterogeneous fibres (purple). Rare collagen fragments interspersed (pink).

For high vs. low COL1A1 and MAF, in Methods, Page 12, Lines 4-7:

“Samples were grouped into COL1A1 high or low based on the expression relative to the median expression of all samples. The melanoma associated fibroblast (MAF) score for each sample was determined by calculating the geometric mean of all genes in a published melanoma associated fibroblast signature²⁷. High and low MAF samples were classified based on the median signature score.”

4. Above concerns should also be considered in the survival data shown from Extended Data Figure 4.

Response 4

Yes, these data are now described in the methods and key legends (see response 1 to Reviewer 4).

REVIEWERS' COMMENTS

Reviewer #1, also commented on behalf of Reviewer #3(Remarks to the Author):

The authors have done an outstanding job of addressing the reviewer comments, and I have no further comments.

Reviewer #2 (Remarks to the Author):

The authors have adequately addressed the reviewers' comments.

Reviewer #4 (Remarks to the Author):

All my comments have been addressed.